# Identification and Functional Analysis of Drought-Responsive Long Noncoding RNAs in Maize Roots

**DOI:** 10.3390/ijms242015039

**Published:** 2023-10-10

**Authors:** Xin Tang, Qimeng Li, Xiaoju Feng, Bo Yang, Xiu Zhong, Yang Zhou, Qi Wang, Yan Mao, Wubin Xie, Tianhong Liu, Qi Tang, Wei Guo, Fengkai Wu, Xuanjun Feng, Qingjun Wang, Yanli Lu, Jie Xu

**Affiliations:** 1Maize Research Institute, Sichuan Agricultural University, Chengdu 611130, China; tangthreeking@126.com (X.T.); lqm1892022@163.com (Q.L.); fxj2014560631@163.com (X.F.); yb18153990764@163.com (B.Y.); zhong18176445116@163.com (X.Z.); yangzh0717@163.com (Y.Z.); qiwang1993@gmail.com (Q.W.); maoyanplant@gmail.com (Y.M.); chuck00544@163.com (W.X.); 71302@sicau.edu.cn (T.L.); qitang927@gmail.com (Q.T.); 71348@sicau.edu.cn (W.G.); wfk0909@163.com (F.W.); xuanjunfeng@sicau.edu.cn (X.F.); wdqdjm@126.com (Q.W.); 2State Key Laboratory of Crop Gene Exploration and Utilization in Southwest China, Sichuan Agricultural University, Chengdu 611130, China

**Keywords:** maize, drought stress, lncRNAs, epigenetic modification, ceRNA networks

## Abstract

Long noncoding RNAs (lncRNAs) are transcripts with lengths of more than 200 nt and limited protein-coding potential. They were found to play important roles in plant stress responses. In this study, the maize drought-tolerant inbred line AC7643 and drought-sensitive inbred line AC7729/TZSRW, as well as their recombinant inbred lines (RILs) were selected to identify drought-responsive lncRNAs in roots. Compared with non-responsive lncRNAs, drought-responsive lncRNAs had different sequence characteristics in length of genes and number of exons. The ratio of down-regulated lncRNAs induced by drought was significantly higher than that of coding genes; and lncRNAs were more widespread expressed in recombination sites in the RILs. Additionally, by integration of the modifications of DNA 5-methylcytidine (5mC), histones, and RNA N6-methyladenosine (m^6^A), it was found that the enrichment of histone modifications associated with transcriptional activation in the genes generated lncRNAs was lower that coding genes. The lncRNAs-mRNAs co-expression network, containing 15,340 coding genes and 953 lncRNAs, was constructed to investigate the molecular functions of lncRNAs. There are 13 modules found to be associated with survival rate under drought. We found nine SNPs located in lncRNAs among the modules associated with plant survival under drought. In conclusion, we revealed the characteristics of lncRNAs responding to drought in maize roots based on multiomics studies. These findings enrich our understanding of lncRNAs under drought and shed light on the complex regulatory networks that are orchestrated by the noncoding RNAs in response to drought stress.

## 1. Introduction

Long non-coding RNA (lncRNA) is a transcript over 200 nt with limited protein-coding potential or functional small peptides [1]. LncRNA genes are mostly produced by RNA polymerase II (RNAPII) transcription. However, RNA polymerase IV and V have also been reported to transcribe lncRNAs [2]. The structure of lncRNAs is similar to the most typical mRNAs, with 3′ polyadenylated (polyA+ or polyA-) and 5′capped structures added before their splicing [3]. In general, lncRNAs can be divided into different categories, including long intergenic non-coding RNAs (lincRNAs), intronic ncRNAs (incRNAs), natural antisense transcripts (NATs), and divergent lncRNA [4,5,6]. The expression of lncRNAs is usually very low and has a specific pattern [7,8,9]. 

LncRNAs have often been reported to be associated with drought tolerance of plants. Zhang et al. [10] identified 8449 drought responsive transcripts and 1724 lncRNAs in B73 leaves under different water conditions, and 664 lncRNAs for drought response. The expression of lncRNA TCON_00021861 could repress miR528-3p on YUCCA7 in rice, resulting in increased indole acetic acid (IAA) content and responding to drought stress [11]. In *Arabidopsis*, DROUGHT-INDUCED lncRNA (*DRIR*) is a positive regulator of drought and salt stress responses by modulating the expression of a series of genes involved in the stress response [12]. The mutation of *StCDF1* in potato and the overexpression of its natural antisense transcript lncRNA *StFLORE* improve drought tolerance by reducing water loss from the plants [13]. *GhDAN1* RNA in cotton primarily regulates drought stress-related genes with AAAG motifs in the auxin response pathway, and silencing *GhDAN1* in plants can improve their tolerance to drought stress [14]. In maize, 1769 pairs of NATs that significantly responds to drought stress are identified and generally exhibit an open chromatin configuration [15].

LncRNAs are known to have various molecular functions, such as modulating chromatin structure and the transcription of genes and affecting RNAs splicing, stability, and translation by interacting with DNA, RNA, and protein [16,17,18]. It can interact with DNA and co-transcriptional form RNA–DNA hybrids, which regulate gene transcription. The lncRNA *TCF21* antisense RNA inducing demethylation (TARID), for example, forms a R-loop which is recognized by GADD45A to trigger local DNA demethylation and TCF21 expression [19]. A study indicated that lncRNAs and circRNAs acting as competitive endogenous RNAs (ceRNAs) have the potential to regulate the expression of osa-miR156l -5p (related to yield), playing important roles in the growth and development of rice [20]. In *Arabidopsis*, lncRNA *IPS1* is induced in a low-phosphorus environment, thereby protecting genes involved in phosphorus homeostasis from miR-399 [21]. LncRNA-XLOC_057324 plays a role in reproductive development with miR160/miR64 through ceRNAs regulation [22]. Moreover, lncRNAs can regulate chromatin structure of genes to bind proteins. For example, lncRNA *CCAT1-L* plays a role in MYC transcriptional regulation and promotes long-range chromatin looping [23]. 

LncRNAs are essential for epigenetic regulation, as lncRNAs modulate the expression of genes through epigenetic mechanisms [24]. They can act as a switch to control gene transcription, by recruiting chromatin-modifying enzymes for DNA or targeting histones to create a compact chromatin structure that blocks access to transcriptional machinery. Furthermore, lncRNAs have also been found to bind directly to specific epigenetic marks, and thereby affect transcription [25]. For example, *LncPRESS1* keeps the active-gene H3 acetylated at Lys56 (H3K56ac) and Lys9 (H3K9ac) modifications as its target genes by recruiting the protein SIRT6 [26]. Epigenetic modifications, such as DNA methylation, histone modification, and RNA methylation, are also closely associated with drought stress [27,28]. Understanding the role of histone modifications in plant responses to drought stress could provide valuable information for increasing drought tolerance and improving crop yields. The exact mechanism of how epigenetic modifications regulate gene expression under drought conditions remains unclear, but researches have revealed that lncRNAs could interact with the writers, readers and erasers of histones [25,29] and also serve as molecular scaffolds of specific chromatin-regulatory complexes, combining and integrating the molecular functions of multiple histone modifications, thereby fine-tuning both the lower- and higher-order chromatin structure of a given target locus [30,31]. 

LncRNAs-mediated chromatin modifications have been reported in plants [32]. The lncRNAs, COLD-ASSISTED INTRONIC NONCODING RNA (COLDAIR) and COOL-INDUCED ANTISENSE INTRAGENIC RNA (COOLAIR), inhibit the *FLOWERING LOCUS* (*FLC*) gene through the lncR2Epi pathway, thereby regulating the flowering period of *Arabidopsis thaliana* under cold stress [33]. In maize, the DNAs methylation of the *ZmNAC111* promoter could inhibit the expression of *ZmNAC111*, resulting in upregulated expression of drought-responsive genes, such as *ZmNCED3*, *ZmRAB18* and *ZmRD17* [34]. Histone modification patterns are associated with specific transcriptional processes. For example, methylation of histone 3 lysine 4 (H3K4me) and histone 3 lysine 36 (H3K36me) are associated with gene activation while deacetylation of histone 3 (H3ac), methylation of histone 3 lysine 9 (H3K9me), and histone 3 lysine 27 (H3K27me) are usually associated with transcriptional inhibition [35]. The downregulation of H3K27me3 and upregulation of H3K4me3 at the *FLC* locus are controlled by exogenously overexpressing COLDAIR in *Arabidopsis* [36]. *IncRNA4* from AGAMOUS (AG) RNAi lines shows decreased H3K27me3 levels of AG mRNA [37]. DRIR is a nucleus-localized lncRNA responding to drought stress via regulating transcription factors (*NAC3* and *WRKY8*) and ABA signal genes (*ABI5*, *RD29A*, and *RD29B*) [12]. In addition, methylation of N^6^-methyladenosine (m^6^A) has been found to be a widely distributed eukaryotic RNAs modification. Recent research showed that m^6^A modification is not only ubiquitous in mRNAs, but also widespread in ncRNAs [38]. M^6^A modification exerts multiple functions on lncRNAs, as it regulates lncRNAs stability [39], subcellular distribution [40], structure [41], and gene transcription [42]. For example, METTL3 increased the stability of *lncMALAT1* by enhancing m^6^A modification [43]. Also, m^6^A positively regulates *lncRP11* expression in colorectal cancer cell [40]. However, few studies revealed the involving mechanism between m^6^A modification and lncRNAs. 

Maize is an important crop that is often limited by drought, resulting in reduced yield. To understand the genetic basis of drought tolerance, it is essential to study the varying drought tolerance among different maize genotypes. In this study, we used the maize roots of drought tolerant related recombinant lines with distinct drought sensitivity. LncRNAs involved in drought response were identified based on stranded-specific RNAs sequencing data. The pattern of lncRNAs expression in drought stress was investigated and the regulatory relationships by the modifications of DNAs, histones, and RNA m^6^A were explored using the sequencing data of methylated DNA immunoprecipitation sequencing (MeDIP-Seq), chromatin immunoprecipitation sequencing (ChIP-Seq) and N6-methyladenosine methylated RNA immunoprecipitation sequencing (MeRIP-Seq) from the aforementioned inbred lines. Moreover, the co-expression and ceRNAs networks were constructed to demonstrate the significant role of lncRNAs in the process of plant’s response to drought stress. In conclusion, this study discussed the formation and function of lncRNAs induced by drought stress and provided a reference for further research on the molecular mechanism of drought tolerance in maize.

## 2. Results

### 2.1. Identification of lncRNAs in Maize Roots at Seedling Stage

A total of 32 transcriptome datasets (after quality control) were aligned to the maize reference genome (AGPv4). The average alignment rate was 96.4% and the average unique alignment rate was 74.07% (The detailed information of each sample is listed in Appendix A). Reads of Ribosome-Nascent-chain-Complex sequence (RNC-seq) were aligned to the transcriptomes that reconstructed through the RNA-seq data of 32 samples. We use Lnc_Finder and CPC2 calculated 29,303 transcripts that lacking coding ability. Next, we filtered transcripts that aligned to protein database using the BLAST program and obtained 10,144 putative transcripts. After eliminated the transcripts with expression values > 0.1 [44,45] in RNC-seq, 4902 putative lncRNAs were obtained. We retained transcripts that length > 200nt and expression abundance > 0.1 in RNA-seq, finally identified 2030 high-confidence lncRNAs. All transcripts originated from 42,427 genes, including 1923 lncRNA genes and 40,504 coding genes. With FPKM ≥ 0.1 as the threshold [7], the number of lncRNAs in each sample was shown in Figure 1A. Comparing the position of lncRNAs in the genome to that of the coding genes, 90.1% (1825) intergenic lncRNAs were identified in this study, which was consistent with the findings of previous studies [7,46]. Other types of lncRNAs accounted for a relatively low abundance. For example, antisense lncRNAs (antisense transcript) accounted for only 4.6% (93) and divergent lncRNAs (within 2 kb upstream of the coding genes and in the opposite direction of transcriptions) accounted for only 3.7% (75), as shown in Figure 1B.

The sequence characteristics of the lncRNAs and coding genes in maize roots were analyzed. Differentially expressed genes (DEGs) under drought stress were calculated through the edgeR. We obtained 1452 (accounting for 75.5% of the identified lncRNAs) differentially expressed lncRNAs and 28,004 (69.1%) differentially expressed coding genes (Appendix A, differentially expressed in at least one inbred line). We performed qRT-PCR validation on a randomly selected set of five lncRNAs in the RIL parental lines under both well-watered (WW) and water stress (WS) conditions. The results demonstrated a relatively high degree of concordance with the ribo-zero sequencing data, specifically confirming the expression of three lncRNAs (XLOC_032450, XLOC_017628, XLOC_028529) as illustrated in Appendix A. Notably, for XLOC_007641, while this lncRNA did not exhibit detectable reads in the AC7729/TZSRW inbred line, qRT-PCR detected its expression. Furthermore, XLOC_027965, closely mirrored our predictions in AC7643 (Appendix A).

Interestingly, the lncRNA genes had fewer exons compared to coding genes. Specifically, 82.3% of lncRNAs were single exon, with a maximum of five exons observed (Figure 1C). Notably, there was a significantly higher proportion of single exon mRNAs that responded to drought compared to those that did not respond (*p* < 0.001, *χ*2 test), However, this trend was not observed in lncRNAs (Figure 1C,D). Moreover, the length of coding genes were significantly longer than that of lncRNA genes (*p* < 0.001, Student’s *t*-test). The length of the lncRNA genes responding to drought was significantly longer than those not responding to drought (*p* < 0.001, Student’s *t*-test), which was inconsistent with the results obtained for coding genes (Figure 1E). These findings indicated that a large number of lncRNAs exhibit distinct characteristics compared to coding genes in maize roots under drought stress. 

### 2.2. Drought Induced lncRNAs in Maize Roots Showed Strong Specificity in Expression

Previous studies have indicated that lncRNAs generally exhibit relatively low expression levels [7]. Similarly, the expression level of lncRNA genes in this study was significantly lower than that of coding genes (*p* = 0.003, Student’s *t*-test). Differential expression analysis showed that the expression levels of lncRNAs and coding genes that responded to drought stress were significantly lower than that did not respond to drought stress (Lnc: *p* < 0.001, Cd: *p* < 0.001, Student’s *t*-test; Figure 2A). 

The expression specificity of lncRNAs and coding genes was revealed through four aspects (Figure 2B–H). Firstly, the proportion of specific expression lncRNAs was 9.31% (179/1923), which was significantly higher than the coding genes (6.61%, 2681/40,504) in inbred lines with different drought sensitivities (D/S) (*p* < 0.001, *χ*^2^ test; Figure 2B). The number of specifically expressed coding genes in drought-tolerant inbred lines was significantly higher than that in drought-sensitive inbred lines (*p* < 0.001, *χ*^2^ test; Figure 2B), whereas no significant difference was observed in lncRNA genes (*p* = 0.663, *χ*^2^ test; Figure 2B). 

Secondly, the impact of different water conditions (WW/WS) on gene expression was analyzed. The number of specifically expressed lncRNAs (113) under WS was significantly higher than those of lncRNAs (73) under WW (*p* = 0.005, *χ*^2^ test; Figure 2C). In contrast, the number of specifically expressed coding genes under drought stress (722) was significantly lower than that under normal conditions (870) (*p* < 0.001, *χ*^2^ test; Figure 2C). The results indicate that drought induces the specific expression of some lncRNAs while inhibiting the expression of some coding genes (Figure 2C). 

Thirdly, specifically expressed lncRNAs accounted for 19.2% (370) of the total in offspring, whereas in parents, only three lncRNAs were specifically expressed. Additionally, the proportion of specifically expressed coding genes was 8.1% (3270), which was significantly lower than that of lncRNAs in offspring (*p* < 0.001, *χ*^2^ test; Figure 2D). In the 16 inbred lines with extreme drought-tolerant or drought-sensitive characteristics, a total of 1320 recombinant fragments were identified using the genotyping-by-sequencing (GBS) data [47]. To explore the origin of lncRNA specifically expressed in the offspring, we explored the lncRNA is derived from recombination by detecting the overlap of lncRNAs genes and the recombination site. These fragments consisted of 310 lncRNAs and 6092 coding genes. Among the genes specifically expressed at the recombination sites in the offspring, there was a significantly higher proportion of lncRNAs (17.4%) compared to coding genes (7.9%, *p* < 0.001, χ^2^ test). This indicates that lncRNAs have a higher tendency to produce specifically expressed genes at recombination sites in the offspring compared to coding genes. Furthermore, when comparing the number of samples with recombination events, it was observed that the proportion of lncRNAs detected in multiple samples with recombination events (23.9%) was significantly higher than that of coding genes (18.0%, *p* < 0.05, χ^2^ test; Figure 2E). This suggests that regions of the genome with high recombination rates in the recombined inbred lines exhibited increased expression of lncRNAs compared to regions with low recombination rates. 

Fourthly, the Shannon entropy of each gene were calculated to evaluate the expression specificity of lncRNAs. The Shannon entropy value closer to 0 indicates that gene expression specificity is strong, and the Shannon entropy value closer to log_2_N (where N is the number of samples) indicates that gene expression specificity is weak. For comparison, the Shannon entropy of housekeeping genes is concentrated at 4 (log_2_16), indicating that it is universally expressed in various RILs. The overall level of Shannon entropy of both coding genes and lncRNAs were lower than housekeeping genes. LncRNAs were enriched at a low Shannon entropy, indicating that lncRNAs have strong material-specific characteristics (Figure 2F). Furthermore, the comparison of the sample specificity between coding genes and lncRNAs in response to drought stress revealed that the Shannon entropy of coding genes that did not respond to drought stress was higher than that of coding genes that responded to drought stress. In contrast, the Shannon entropy of lncRNA genes that responded to drought stress was significantly higher than that of lncRNA genes that did not respond to drought stress (Figure 2G). In addition, among the coding genes, the Shannon entropy of the coding genes specifically expressed in drought-tolerant and drought-sensitive inbred lines was not significantly different, whereas the Shannon entropy of lncRNA genes specifically expressed in drought-tolerant inbred lines was lower than those expressed in drought-sensitive inbred lines (Figure 2H). 

### 2.3. LncRNAs in Maize Roots Respond to Drought Stress

The proportion of lncRNA genes expressed in response to drought stress was found to be significantly higher than that of coding genes (*p* < 0.05, *χ*^2^ test; Figure 3A). A comparison of the distribution in the expression fold change of (log_2_ fold change, log_2_FC) lncRNA, coding, and it revealed that the fold change of housekeeping genes was close to 1. Notably, the lncRNAs exhibited a larger range of fold change than coding genes, indicating that drought has a greater impact on lncRNAs (Figure 3B). Further analysis of the up-regulation of lncRNA and coding genes revealed that while the number of up-regulated coding genes accounted for 21.4% of the total number of responses, the up-regulated ratio of lncRNA genes (38.9%) was significantly higher than that of coding genes (*p* < 0.001, *χ*^2^ test). Similarly, the down-regulation ratio of lncRNA genes (50.5%) was also significantly higher than that of coding genes (29.8%). Although the down-regulation ratio was significantly higher than the up-regulation ratio in lncRNA and coding genes (*p* < 0.001, *χ*^2^ test), the number of down-regulated and up-regulated lncRNAs were significantly different from that of coding genes (*p* < 0.001, *χ*^2^ test; Table 1). We further analyzed the up-regulated and down-regulated of specifically expressed lncRNAs in D/S (shown in Appendix A). It was found that the number of up-regulated specifically expressed lncRNAs in drought-tolerance inbred lines was significantly lower than that of in drought-sensitive inbred lines (*p* < 0.001, *χ*^2^ test, Appendix A). While the number of down-regulated specifically expressed lncRNAs was significantly in D higher than that of in S (*p* < 0.001, *χ*^2^ test, Appendix A).

The number of differentially expressed genes (DEGs) in drought-stress response were analyzed between the parent and offspring generations of RILs, as well as between drought-tolerant and drought-sensitive lines. Among the offspring, a vast majority of genes (91.1% or 1323) specifically responded to the stress, while only a small fraction (0.5% or 8) of parent-specific genes responded to the stress. The similar trends were observed in coding genes, with a significantly higher number of specific stress-response genes in the offspring compared to the parents (*p* < 0.001, χ^2^ test; Figure 3C). The proportion of lncRNAs specifically responding to drought stress in the offspring was significantly higher than that of coding genes (*p* < 0.001, χ^2^ test; Figure 3C). Conversely, the proportion of lncRNAs that specifically responding to stress in the parent was significantly lower than that of coding genes (*p* < 0.001, χ^2^ test; Figure 3C). A comparison of the specific stress responses between drought-tolerant and drought-sensitive inbred lines did not reveal significant differences in lncRNAs and coding genes. The proportion of lncRNA genes that specifically responded to stress was significantly higher than that of coding genes in different tolerant inbred lines (drought-tolerant RILs: *p* < 0.05; drought-sensitive RILs: *p* < 0.01, χ^2^ test; Figure 3C). In general, lncRNA genes exhibited higher expression levels and stronger expression specificity under drought stress compared to coding genes, particularly in the RILs. 

To gain insight into the potential molecular functions of lncRNA and coding genes in drought response in maize roots, we investigated the correlations between sequence variations in drought responsive lncRNAs and plant survival rate under drought. We selected an equal number of coding genes with equivalent expression levels as negative controls and compared the number of single nucleotide polymorphisms (SNPs) significantly associated with plant survival rate on these genes. The results revealed a significant enrichment of SNPs associated with plant survival under drought stress in lncRNA genes within the association mapping panel consisting of 368 maize inbred lines [48] (Figure 3D). While this trend was not observed in the trait of maize kernel oil content (Figure 3D).

### 2.4. Differences in Epigenetic Modifications between lncRNA Genes and Coding Genes 

Histone modifications can regulate gene expression through various mechanisms. To investigate how histone modifications affect the expression of lncRNAs, we compared the histone modifications within 1 kb upstream and downstream of lncRNA and coding genes. The levels of various histone modifications such as H3K4me1, H3K4me3, H3K9me3, H3K36me3, H3K9ac and H3K27ac were also compared between drought and well-watered conditions. Genes generate lncRNAs exhibited distinct patterns of histone modifications compared to the coding genes. Histone methylation and acetylation were enriched at the transcription start site (TSS, 5′-end) of genes, but lncRNA genes displayed fewer active chromatin regions than coding genes (Figure 4A–F). Specifically, H3K4me3 modification was enriched at the TSS of lncRNA and coding genes, but the peaks of H3K4me3 modification in lncRNA genes was broader and more dispersed than coding genes (Figure 4A). The 5′-end of the coding genes was enriched with H3K4me3 and H3K9ac modifications, while this trend was not as obvious as in lncRNA genes (Figure 4A,C). LncRNA genes were also enriched in three kinds of histone modifications associated with transcriptional activation (H3K4me3 and H3K9ac; Figure 4A–C). However, lncRNA genes exhibited lower modification level than coding genes, which may contribute to the low expression level of lncRNAs. 

In addition, the modification levels of H3K9me3, H3K4me1, and H3K36me3 in the lncRNA genes and coding genes increased under drought stress (Figure 4B,E,F). Conversely, the levels of H3K4me3 and H3K27ac modifications were down-regulated under drought (Figure 4A,D). It is worth noting that the H3K9ac modification level at the 5′-end of the coding gene was significantly down-regulated, while there was no significant change in lncRNA genes (Figure 4C). Furthermore, the enrichment level of H3K4me1 in the gene body was increased under drought (Figure 4E). The modification levels of H3K4me3 and H3K9ac in lncRNA genes body was lower under drought than normal conditions (Figure 4A,C). H3K36me3 modification is usually associated with transcription activation and alternative splicing. The H3K36me3 modification signal was weak in 5′-end of lncRNA genes regions, which could be caused by the large number of lncRNAs with single exon and associated with less alternative splicing events (Figure 4F). 

On the other hand, the lncRNA genes showed a higher level of DNA methylation than the coding genes (Figure 4G). In addition to analysis DNA-level modifications, we also analysis m^6^A modifications in lncRNAs and mRNAs. Our findings revealed that m^6^A modification levels were increased in both lncRNAs and mRNAs under drought stress (Figure 4H). For mRNAs, as expected, m^6^A were enriched at the 3′-end, near the stop codon regions (Figure 4H). However, for lncRNAs, no obvious m^6^A enrichment signals were observed (Figure 4H).

The epigenetic modifications of lncRNAs and coding genes specifically expressed in inbred lines under different water conditions were also analyzed (Figure 5). The H3K9ac modification of lncRNA genes that specifically expressed in drought-sensitive inbred lines was significantly higher than that specifically expressed in drought-tolerant inbred lines (*p* < 0.05, Student’s *t*-test, Figure 5A and Appendix A). In contrast, this trend was not observed in coding genes with H3K9ac modification (Figure 5B).

### 2.5. Regulatory Networks of the lncRNA–mRNA and lncRNA-miRNA-circRNA/mRNA 

A weighted gene co-expression network analysis (WGCNA) was constructed using 28,004 coding and 1452 lncRNA DEGs. We obtained 63 modules, of which the largest module contained 6561 genes and the smallest module contained 37 genes. The co-expression network contained 15,340 coding and 953 lncRNA genes. Next, we investigated the correlations between 23 phenotype traits and modules and obtained 23 modules (correlation coefficient > 2 or <−2 and *p* < 0.05) for subsequent analysis (Figure 6). The number of genes included in the 23 modules was shown in Appendix A, which consists of 11,495 coding and 558 lncRNA genes. 

Among the 23 modules, 13 modules were found to be associated with survival rate under drought stress, with eight modules showing a positive correlation with survival rate. The MEmidnightblue module exhibited a significant negative correlation (correlation coefficients: −0.65, correlation test, *p* < 0.001) with the survival rate at seedling stage. Conversely, the MEyellowgreen module was significantly positive correlation (correlation coefficients: 0.5, correlation test, *p* < 0.001) with the survival rate. In addition, the MEturquoise, MEcyan, and MEgreen modules showed positive correlation with five root system traits (root dry weight, root length, root surface area, root volume, and root branch number) and plant height phenotypes, while MEblue and MEpink showed a significant negative correlation (Figure 6). Notably, these modules that respond to drought stress might play important roles in maize roots. 

We investigated the correlation coefficients between modules membership (MM) and gene significance (GS) for survival rate under drought and identified 14 modules that exhibited significant correlations (correlation coefficients > 0.3 or <−0.3 and *p* < 0.05, Figure 7). Among these modules, the MEyellowgreen module showed the highest correlation coefficient of 0.83 (Figure 7A), the MEcyan and MEbrown modules reached correlation coefficients of 0.79 and −0.70, respectively (Figure 7B,C). The higher the correlation between the MM and the GS, the stronger the positive correlation between the module and survival rate. Although the MEturquoise module containing the largest number of genes, its correlation coefficient reached −0.35 (Figure 7D). This suggests that the genes within this module are associated with the survival rate of maize under drought. 

The SNPs associated the drought-tolerant traits identified by the genome-wide association analysis (GWAS) were used to estimate the potential functions of lncRNAs. A number of 192 significant SNPs located within 35 lncRNA genes were identified. Among them, 11 SNPs were found to be associated with survival rate under drought. The comparisons of plants survival rate under drought between different genotypes were shown in Figure 8. Significant differences in survival rate were observed among maize lines with different genotypes (Appendix A).

Previous studies have demonstrated that lncRNAs could act as ceRNAs and participate in various biological processes [49]. In our study, we integrated lncRNAs, circRNAs and mRNAs to construct a ceRNAs regulatory network with miRNAs serving as the central regulatory molecules. The results included 169 lncRNAs, and 425 differentially expressed circRNAs and 7848 mRNAs, collectively form a complex regulatory network with a number of 18,569 regulatory relationships (Figure 9). Different types of RNAs were found to interact with at least one miRNA in the network. For example, lncRNA XLOC_020662 could binding zma-mir408b-5p; this miRNA, in turn, could also bind to three circRNAs and 30 mRNAs, especially one of which was circRNA-2_3953473_3954253 and lncRNA XLOC_014287 also interacted with zma-mir399a-5p. The results indicated that lncRNAs could interact with miRNAs to modulate the expression of downstream genes, thereby playing roles in the response to drought stress.

## 3. Discussion

LncRNAs are transcripts longer than 200 nt and lack significant protein coding potential [17,50]. With the emergence of high-throughput sequencing technologies and the development of bioinformatics, numerous programs have been developed to discern the distinctions between coding and long noncoding transcripts. Currently, most programs for lncRNA identifications are alignment-based tools, with differences in the analysis of lncRNA coding ability potential. The machine learning based methods CPC and CNCI were used in previous studies to identify lncRNAs [51,52]. Li et al. employed CPC2 software (version 0.1) and alignment with known protein databases to identify the coding ability of the maize lncRNAs [7]. Compared with previous studies, our study took a more stringent approach to identify the lncRNAs. We utilized two softwares for predicting lncRNAs and also incorporated RNC-seq data to assess the coding ability of each gene. RNC-seq provides valuable information about the RNAs that bind to ribosomes and participate in translation [53]. Only 4902 transcripts remained combined with RNC-seq data. Thus, compared with the existing research processes for lncRNAs identification, the method we employed in our research was more reliable. 

The overall expression level of lncRNAs in this study was found to be low, which might be associated with the low histone modification level at their promoter regions [7,8,54,55]. The histone modification level of lncRNAs in our study confirmed this trend. Specifically, the enrichment levels of histone modifications associated with transcriptional activation (H3K4me3, and H3K9ac) in lncRNA genes were significantly lower than that of the coding genes (Figure 4A,C).

This not only indicates the overall low expression level of lncRNAs, but also highlights their strong tissue specificity [7]. Previous studies utilized Shannon entropy to evaluate the specificity of gene expression among different maize inbred lines. Li calculated the Shannon entropy of lncRNAs and found that it was tissue specific [7,56]. We followed this method and found that lncRNA genes exhibited stronger material specificity than coding genes. Moreover, among the lncRNA genes, those that did not respond to drought stress showed even stronger material specificity compared to the ones that responded to stress (Figure 2G). These findings suggest that lncRNAs have a tendency to stably respond to drought.

Through the calculation of offspring recombination events, we found that the recombination rate on lncRNA genes was similar to that of the coding genes. However, not all recombination events resulted in the specific expression of lncRNA in the offspring (Figure 2E). The proportion of the recombination events converted into offspring-specific expression in the lncRNA genes (17.4%) was significantly higher than that in coding genes (7.9%). This suggests that more specifically expressed lncRNAs derived from recombination sites. 

Although lncRNA cannot encode a protein, it can respond to drought stress through various regulatory pathways [55]. The number of drought-responsive lncRNAs was significantly higher than that of coding genes, and the DEGs of lncRNAs exhibited a broader distribution under stress. Interestingly, the expression of DEGs (both up-regulation and down-regulation) was significantly lower than that those that respond to drought, indicating that lncRNA genes were more sensitive to respond drought (Figure 2A). The predominance of down-regulated genes under drought can be attributed to the plant’s adaptive response mechanisms. Plants often reduce the expression of non-essential genes to conserve energy and resources for survival under stressed conditions. Conversely, up-regulation genes typically involves in the plant’s defense and adaptation. As shown in Table 1, the down-regulation rate is significantly higher than the up-regulation rate in both lncRNAs and coding genes.

Histones—the basic unit of nucleosomes—are rich in lysine at the amino terminus and can undergo various modifications (such as methylation, acetylation, phosphorylation, and ubiquitination) that alter the chromatin state to regulate gene expression [16]. Epigenetic mechanisms play a crucial role in regulating genes’ response to drought stress at both the transcriptional and post-transcriptional levels by modifying the chromatin state of genes. Examples from other studies have also demonstrated the importance of histone modifications in response to drought stress. For instance, in Arabidopsis, an increase in the H3K4me3 modification level in the NCED3 gene resulted in a significant increase in NCED3 gene expression under drought conditions [57]. Similarly, under drought stress, the H3K4me3 and H3K9ac modifications in the RD29A promoter were significantly upregulated, leading to increased gene expression [58]. In *Populus pilosa*, the binding of AREB1 to the ABRE motif of the NAC gene enhanced the H3K9ac modification level in NAC, activating the expression of PtrNAC gene and improving drought resistance [59]. In this study, we observed that H3K9ac modification level of the lncRNA genes in drought-sensitive inbred lines was higher than that of drought-tolerant inbred lines (Figure 5A). However, this trend was not found in coding genes (Figure 5B). The specific expression lncRNAs in drought-sensitive inbred lines may be due to the increased H3K9ac modification level, which may have a negative impact on drought resistance in sensitive inbred lines. 

To identify candidate lncRNA genes associated with drought traits, we constructed a co-expression network based on the DEGs obtained under drought conditions, resulting in 16 modules significantly associated with drought traits (Figure 6). Fortunately, we found a significantly up-regulated WRKY (XLOC_030732) in drought-sensitive inbred line in MEbrown significantly associated with drought survival. Combined with the GWAS results, six candidate lncRNA genes were identified (Appendix A). It revealed that nine SNPs located in six genes could differentiate from genotypes associated with survival rates under drought (Figure 8). These SNPs are excellent candidates for downstream experimental studies and further research.

MiRNAs, as the center of the co-expression network, facilitating the communication between lncRNAs, circRNAs and mRNAs. In this study, we found that lncRNA-XLOC_020662 can adsorb zma-miR408b-5p. The miR408 families were significantly down-regulated in wheat roots but up-regulated in wheat leaves under drought stress [60]. The high expression of miR408 improved tolerance to salinity and cold stress, but enhanced sensitivity to drought stress [61]. It has been reported that the down-regulation expression of miR408 expression in rice affects DEAD-box helicases and play a critical response drought stress [62]. We found a zma-miR399a-5p that cooperates with circRNA-2_3953473_3954253 and lncRNA-XLOC_014287. The miR399 family members in *Haberlea rhodopensis* were down-regulated during dehydration [63]. MiR399 was down-regulated in B73 shoot apical meristems under drought [64]. The ceRNA networks provides rich resources for studying stress resistance. These findings suggest that lncRNAs and circRNAs may collectively regulate miRNAs in response to drought stress, providing valuable insights into stress resistance mechanisms.

## 4. Conclusions

This study sheds light on the characteristics of lncRNAs expression and regulation in maize roots in response to drought stress. It was discovered that lncRNAs in maize roots were more preferentially expressed at the recombination sites compared to coding genes. LncRNAs specifically expressed in drought-sensitive inbred lines showed higher H3K9ac modification levels than those specifically expressed in drought-tolerant inbred lines. By constructing a co-expression network, nine SNPs located in lncRNA genes were identified, showing significant associations with survival rate under drought conditions. The ceRNA networks revealed multiple regulatory relationships involved in the response to drought stress. Through a multiomics perspective, this study further explores the biological significance of lncRNAs in maize roots for their role in responding to drought stress. 

## 5. Materials and Methods 

The inbred lines used in this study were the same as our previous research [65]. The maize drought-tolerant inbred line AC7643, the drought-sensitive inbred line AC7729/TZSRW, and 14 recombinant inbred lines (RILs) were chosen based on the leaves death rate under drought stress [47]. The 14 RILs were divided into drought-tolerant and drought-sensitive RILs based on their leaves death rate under drought. The drought-tolerant RILs used were RIL231, RIL131, RIL155, RIL203, RIL142, RIL208, and RIL165; the drought-sensitive RILs used were RIL126, RIL64, RIL226, RIL27, RIL47, RIL8, and RIL166. Inbred lines grown in nutrient solution (4 mM Ca(NO_3_)_2_·4H_2_O, 6 mM KNO_3_, 1 mM NH_4_H_2_PO_4_, 2 mM MgSO_4_·7H_2_O, 0.1 mM Na·Fe·EDTA, 46 μM H_3_BO_3_, 9.146 μM MnCl_2_·4H_2_O, 0.76 μM ZnSO_4_·7H_2_O, 0.32 μM CuSO_4_·5H_2_O, and 0.016 μM (NH_4_)6MO_7_O_24_·4H_2_O). The plants at the five-leaf stage were then treated with 20% (*w/v*) polyethylene glycol PEG 6000 (Sigma-Aldrich, St. Louis, MO, USA) for 24 h (WS), whereas the control group was grown in the normal nutrient solution (WW). Roots from three plants per inbred line were harvested, pooled, flash frozen in liquid nitrogen, and stored at −80 °C. 

### 5.1. Libraries Construction and RNA Sequencing

RNA strand-specific library constructed refer to Xu et al. [15]. RNAs were isolated from two independent replicates of each line under two water conditions for strand-specific library construction and sequencing. The first-strand of cDNA was synthesized using random hexamer primer and second-strand was synthesized with dUTP instead of dTTP. Double-strand cDNA fragments were purified and ligated with adaptors. Libraries were high-throughput sequenced at Beijing Genomics Institute (Shenzhen, China).

The ribosome nascent-chain complex (RNC) was extracted according to method [66]. Then, RNA extracted from RNC using RNAiso Plus reagent, the RNC-RNA library construction and sequencing by BGI genomic Co., Ltd. (Shenzhen, China), it same as RNA-seq. 

### 5.2. Identification of LncRNAs

A total of 32 transcriptome datasets (after quality control) aligned to the maize reference genome (AGPv4), the average alignment rate was 96.4% and the average unique alignment rate was 74.07%. Reads of RNC-seq were aligned to the transcriptomes that reconstructed through the RNA-seq data of 32 samples. The RNC-seq detects only the translating RNAs. We use Lnc_Finder [67] and CPC2 [68] calculated 29,303 transcripts that lacking coding ability. Next, we filtered transcripts that aligned to the protein database https://download.maizegdb.org/Zm-B73-REFERENCE-GRAMENE-4.0/Zm-B73-REFERENCE-GRAMENE-4.0_Zm00001d.2.protein.fa.gz (accessed on 3 May 2019) using the BLAST program (BLAST 2.3.0+) The alignment was considered as successful when the following three conditions were simultaneously satisfied: alignment rate ≥ 65, alignment length > 30 amino acid, and E-value ≤ 1 × 10^−9^. And 10,144 putative transcripts were obtained. After eliminated the transcripts with expression values > 0.1 [44,45] in RNC-seq, 4902 putative lncRNAs were obtained. We retained transcripts that length > 200 nt and expression abundance > 0.1 in RNA-seq, finally identified 2030 high-confidence lncRNAs. All transcripts originated from 42,427 genes, including 1923 lncRNA genes and 40,504 coding genes. The lncRNAs obtained through the identification process were considered as highly reliable for downstream analysis.

### 5.3. Transcriptome Annotation and Expression Calculation

The raw sequencing data was subjected to sequence quality evaluation, and the unqualified sequences, including sequences containing the adapters, base N, and sequences with a quality value Q ≤ 10, were removed to obtain effective sequences or clean reads. Tophat2 (ver.2.1.0) [69] was used to align the clean reads to maize B73 AGPv4 genome. Stringtie [70] (ver. 1.3.5) was used to generate the GTF files of the different 32 samples. Subsequently, Cuffmerge [71] was used to merge the gene transfer format (GTF) files of each material, resulting in the final transcriptome annotation file. The clean reads were re-align to the maize B73 AGPv4 genome using STAR (ver. 2.7.1a) with --outSAMunmapped Within --outFilterMultimapNmax 10 [72]. Then, cufflinks [71] used to calculated FPKM (fragments per kilobase million), with merged GTF files from the previous steps as a guide.

### 5.4. qRT-PCR

Total RNA was extracted from roots using the RNAiso Plus reagent (Takara Bio Inc., Otsu, Shiga, Japan) according to the user’s manual. RNA was reverse transcribed by HiScript II 1st Strand cDNA Synthesis Kit (R212-01, Vazyme, Nanjing, China) and qRT-PCR using ChamQ SYBR qPCR Master Mix (Q411-01, Vazyme, Nanjing, China). Actin and GAPDH were used as references. The sequences of primer shown in Appendix A.

### 5.5. Differential Expression Analysis

To compare the differences in gene expression under different water conditions, the “featurecount” function in R language (version 3.2.3) package Rsubread [73] package was used to calculate the gene reads; and the R package edgeR [74] was used to investigate the differential expression between two replicates. Likelihood ratio tests were used to analyze differential gene expression, and genes with a corrected *p* value < 0.05 and|log_2_ (fold change)| ≥ 1 were considered as differentially expressed genes (DEGs).

### 5.6. LncRNA Expression Pattern Analysis

The R language package edgeR was used to identify lncRNAs that significantly responded to stress, and the up-regulated and down-regulated genes were identified in the parent/offspring and drought tolerance groups. The distribution pattern of offspring-specific lncRNAs in the whole genome was determined, and we analyzed whether these lncRNAs were enriched at the recombination sites. Additionally, we examined whether the expression and response patterns of lncRNA were consistent with those of their parents genes. Shannon entropy (R language scripts) was used to evaluate the expression specificity of genes in all the maize lines studied. Shannon entropy was calculated as follows:$$H\left(X\right)=-\sum_{x}P\left(x\right){log}_{2}\;[P\left(x\right)]$$
where *H*(*X*) is the Shannon entropy, *x* is the gene expressed in each sample, and *P*(*x*) is the relative expression in each sample. 

The genotypes of the RIL population and the maize natural population with 368 different maize inbred lines are provided by Prof. Xuecai Zhang [75] (downloaded from The iPlant Collaborative https://cyverse.org/, accessed on 19 November 2014) and Prof. Jianbing Yan [76] (downloaded from http://www.maizego.org/, accessed on 16 December 2012), respectively. We utilized the Genome-Wide Association Study (GWAS) using the plant survival rate under drought [77] and kernel oil content from published studies [76]. SNPs that exhibited significant associations with survival rate and kernel oil content were filtered and then the SNP numbers overlapped within the lncRNA locus and protein-coding genes were calculated to evaluate the potential functions of lncRNAs and genes.

### 5.7. MeDIP-seq

MeDIP sequencing data from the roots of two extreme drought-tolerant inbred lines (AC7643 and RIL208) and two extreme drought-sensitive inbred lines (AC7729/TZSRW and RIL64). The data were generated under WW and WS conditions in our previous study and were obtained from the NCBI SRA database (accession number: SRP063383) [15]. 

High-quality reads were mapped to the maize B73 reference v4 (Refv4) genome using bowtie2 (ver.2.2.9) with default parameters and the best-matched reads were used in the downstream analysis. The method refer to our previous study [47].

### 5.8. M^6^A-seq

In this study, m^6^A immunoprecipitation was investigated according to Dominissini’s method [78]. A volume of 5 mg of total RNA without rRNA was broken into approximately 100 bp fragments using a Qsonica Q800R3 sonicator (Qsonica LLC, Newton, MA, USA). Then, the sample was divided into two parts: one part was incubated with an m^6^A antibody (No. 202 003, Synaptic Systems, Goettingen, Germany), and the other part was incubated with an IgG antibody (ab172730, Abcam, Shanghai, China). Immunomagnetic Protein A beads (Repligen, Waltham, MA, USA) were used to enrich the RNA fragments containing m^6^A methylation. The fragments were enriched and used for library construction. The libraries were sequenced using the Illumina HiSeqTM 4000 sequencer (BGI genomic Co., Ltd., Shenzhen, China). The effective sequences were extracted from the raw sequencing data after removing adapters, low-quality reads, and sequencing contaminants. 

### 5.9. ChIP-seq

Three-gram leaf samples from maize seeding were cross-linked and fixed with 1% formaldehyde for 15 min, and then 1.25 M glycine was added to terminate the cross-linked reaction. After the cross-linked samples was completely grounded into powder with liquid nitrogen and the nucleoprotein was extracted, the genomic DNA was broken into 200–500 bp fragments using the Qsonica Q800R3 sonicator. A part of the fragmented samples incubated without antibody was used as an input. Additionally, histone modification-specific antibodies (H3K4me1, H3K4me3, H3K9ac, H3K9me3, H3K27ac, H3K27me3, and H3K36me3) and Protein A/G magnetic agarose beads (No. 78609, Thermo Scientific, Waltham, MA, USA) were added to the remaining sample, and the mixture was incubated for 2 h at 4 °C. Subsequently, the protein-DNA complexes from the immunoprecipitation were collected using a magnetic rack and de-crosslinked using RNase A and proteinase K for 8 h. DNA was precipitated using sodium acetate and ethanol, dried, and dissolved in 20 μL ddH_2_O to construct a library for next-generation sequencing. 

### 5.10. Phenotypic Data Collection

Different drought resistant materials were planted in 30 × 30 pots, with five seedings planted in sandy soil (nutrient soil: sand = 1:3). A treatment and a control group were set up. The treatment group stopped watering when the maize grew to three-leaf stage, while the control group maintained a normal amount of watering. After 40 days of drought treatment, the dry weight and fresh weight of aboveground, plant height, leaves death rate, SPAD, and leaf mortality rate, SPAD value, and plant survival rate were measured. The experiment has two replicates. We use the phenotypic data as above describe to construction the WGCNA network.

### 5.11. Construction of the lncRNA Based Maize Root Co-Expression Network in Response to Drought Stress 

The R language package WGCNA [79] was used to construct a co-expression network of lncRNA and coding genes in response to stress. The correlation between the characteristic value of each module and the phenotype was analyzed, and the modules that were significantly associated with the same phenotype under the two repetitions were selected for further analysis. 

### 5.12. Construction of the lncRNA/circRNA-miRNA-mRNA Network 

We constructed miRNA-lncRNA/circRNA and miRNA-mRNA regulatory relationships with miRNA as a bridge refer to previous research [80]. By combining miRNA-lncRNA/circRNA and miRNA-mRNA regulatory relationships the lncRNA/circRNA-miRNA-mRNA regulatory network was then established using Cytoscape software. MiRNAs data derived from previous research (PRJNA294848, PRJNA816639) [65], circRNAs data will be published in another study.

## Figures and Tables

**Figure 1 ijms-24-15039-f001:**
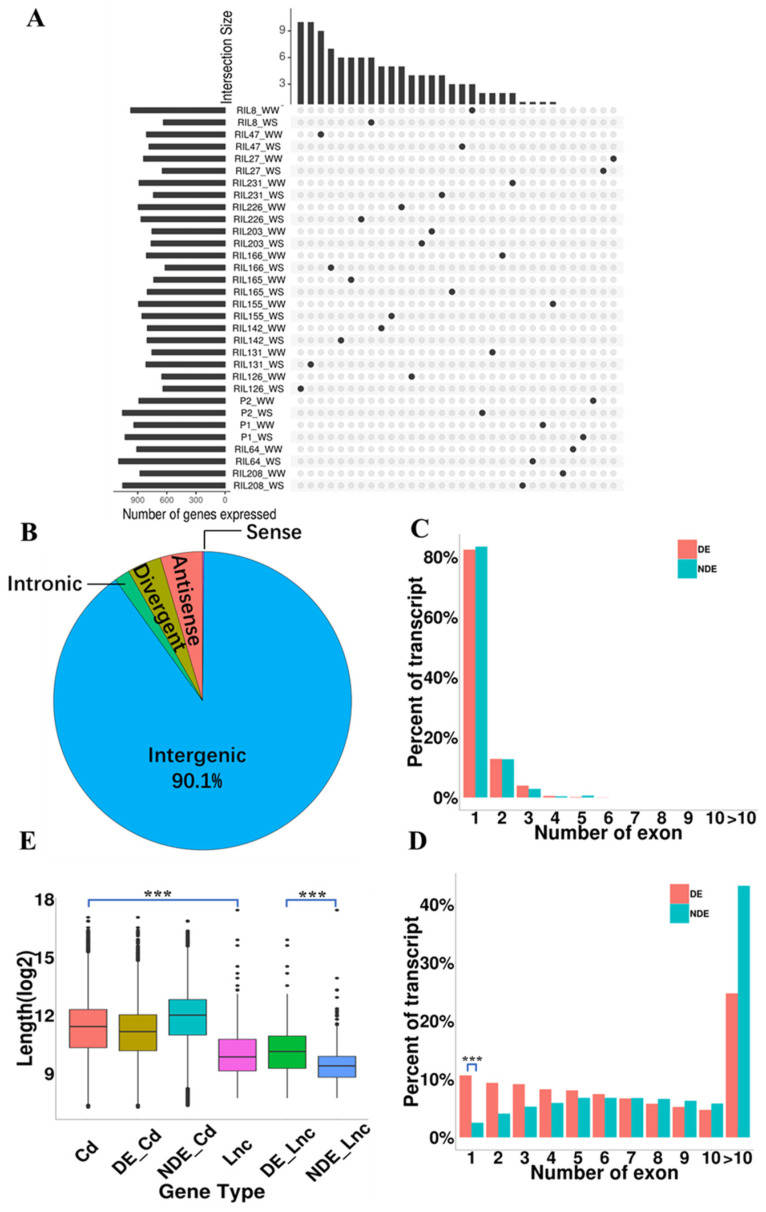
Identification and characterization of lncRNAs in maize roots. (**A**) Distribution of lncRNAs in the RILs. This figure consists of two histograms and a dot matrix diagram. The left histogram represents the total number of lncRNAs detected in each inbred line, while the upper histogram represents the number of specifically expressed lncRNAs in the inbred lines where the black dots are located. P1 and P2 are maize inbred line AC7643 and AC7729/TZSRW, respectively. (**B**) Classification of lncRNAs relative to the different genomic regions. Percentage of the number of exons in (**C**) lncRNAs and (**D**) mRNAs. DE, differential expression; NDE, no differential expression. (**E**) Length distribution of different kinds of genes. Cd, coding genes; DE_Cd, differential expression of coding genes; NDE_Cd, no differential expression of coding genes; Lnc, lncRNA genes; DE_Lnc, differential expression of lncRNA genes. NDE_Lnc, lncRNAs expressed with no difference. ***, *p* < 0.001.

**Figure 2 ijms-24-15039-f002:**
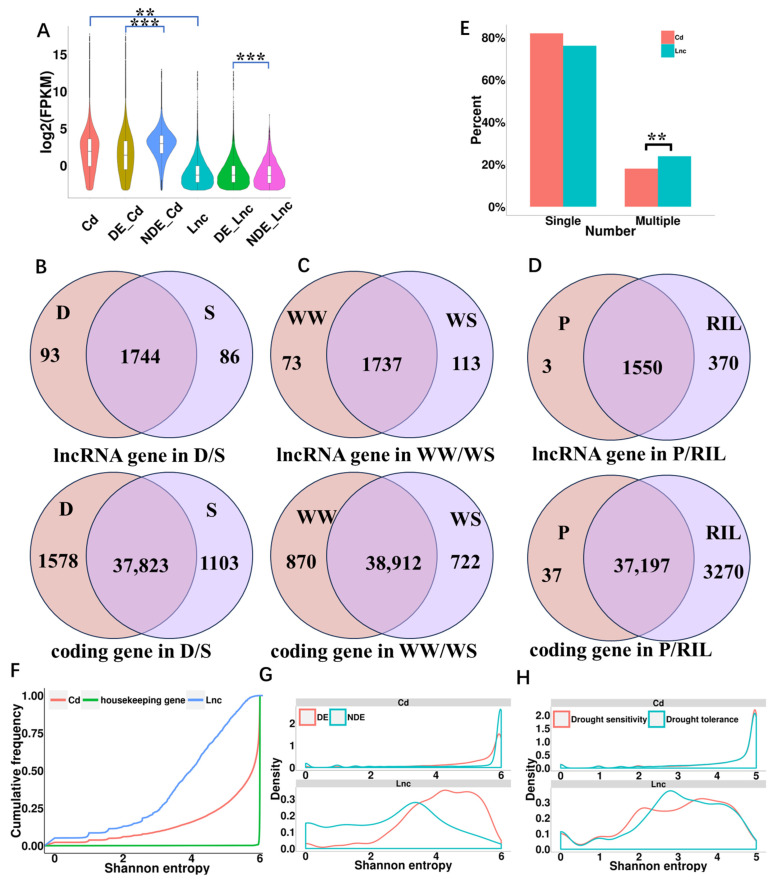
Expression characteristics of lncRNAs in maize roots. (**A**) The expression levels of genes. Cd, coding genes; DE_Cd, differential expression of coding genes; NDE_Cd, no differential expression of coding genes; Lnc, lncRNA genes; DE_Lnc, differential expression of lncRNA genes; NDE_Lnc, no differential expression of lncRNAs. (**B**–**D**) The number of expressed lncRNA genes and coding genes between (**B**) inbred lines of varying drought tolerance, (**C**) different environmental conditions, and (**D**) different generations. D, drought-tolerant; S, drought-sensitive; WW, well-watered; WS, water stress; P, parent inbred line; RIL, recombinant inbred line. (**E**) Proportion of RILs with recombination events. Single, the recombination events were detected in only one sample. Multiple recombination events were detected in two or more samples (**F**) Shannon entropy density distribution of different genes. (**G**,**H**) Shannon entropy density distribution of (**G**) differentially expressed genes and (**H**) RILs of varying drought tolerance. Cd, coding genes. Lnc, lncRNAs. **, *p* < 0.01; ***, *p* < 0.001.

**Figure 3 ijms-24-15039-f003:**
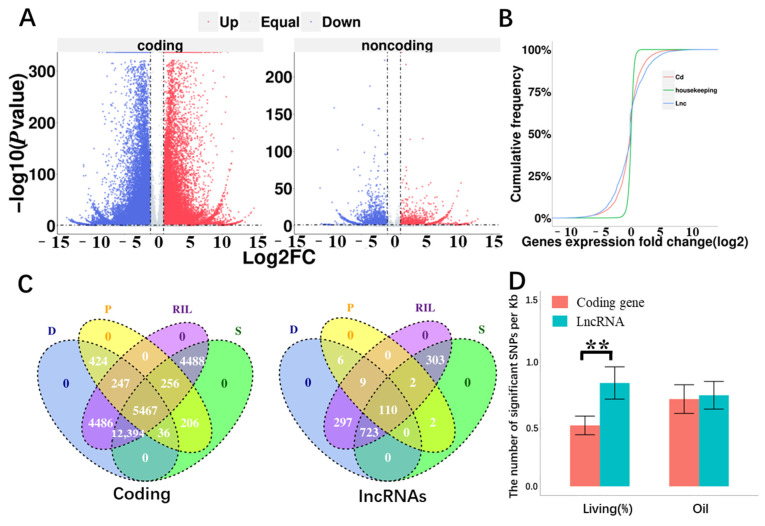
Drought-response lncRNAs in maize roots. (**A**) Volcano plot displayed differentially expressed coding genes (**left**) and lncRNAs (**right**) in maize roots under drought. Down-regulated genes were marked in blue; up-regulated genes were marked in red; and non-differentially expressed genes were marked in gray. The *X*-axis represents log_2_(FC) and the *Y*-axis represents the -log_10_(*p* value). (**B**) Cumulative probability density distribution of gene expression fold change. Green represents housekeeping genes; red represents coding genes; and blue represents lncRNA genes. (**C**) The Venn diagram of genes responses to drought stress. D: drought-tolerant. S: drought-sensitive P: parent inbred line. RIL: recombinant inbred line. (**D**) The average number of SNPs (per kb) associated with survival rate (living %) and oil (kernel oil content) in maize association population for lncRNA and coding gene. Living (%) represents the survival rate under drought stress. **, Student’s *t*-test, *p* < 0.01.

**Figure 4 ijms-24-15039-f004:**
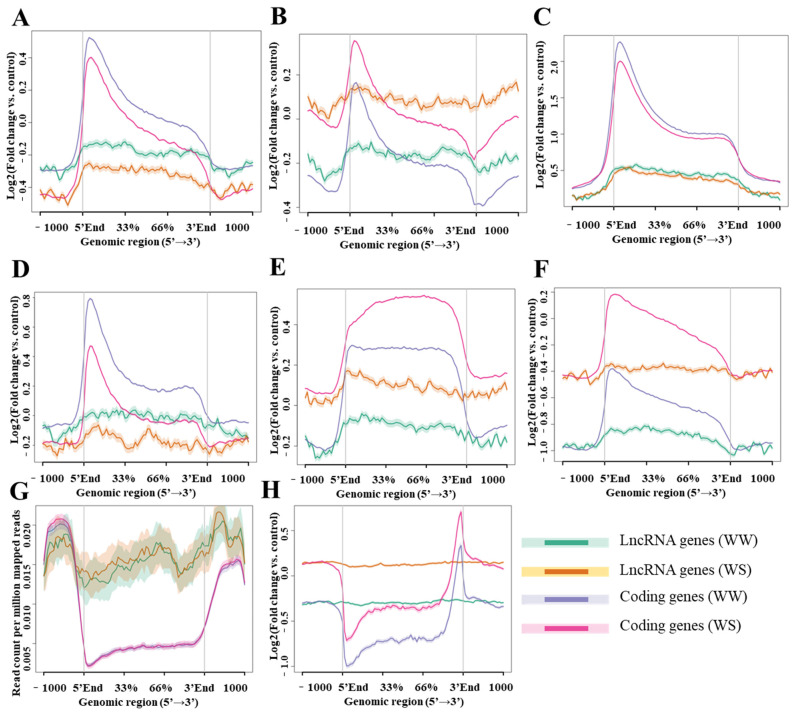
Epigenetic modifications of lncRNA genes and coding genes under different water conditions. (**A**–**F**) Histone modifications: (**A**) H3K4me3, (**B**) H3K9me3, (**C**) H3K9ac, (**D**)H3K27ac, (**E**) H3K4me1, (**F**) H3K36me3, (**G**) DNA methylation, and (**H**) RNA m^6^A modification. The *X*-axis represents the gene regions, and the *Y*-axis represents the enrichment level.

**Figure 5 ijms-24-15039-f005:**
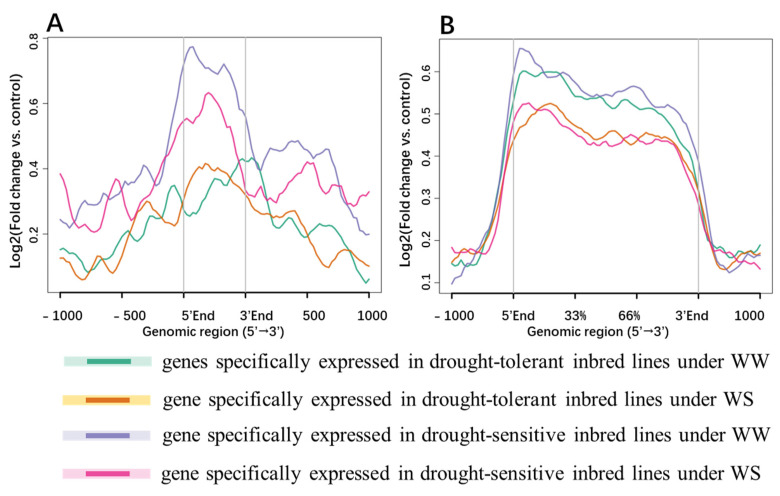
H3K9ac modifications level of lncRNA genes (**A**) and coding genes (**B**) specifically expressed in different drought resistant inbred lines under different water condition. WW, well-watered; WS, water-stressed.

**Figure 6 ijms-24-15039-f006:**
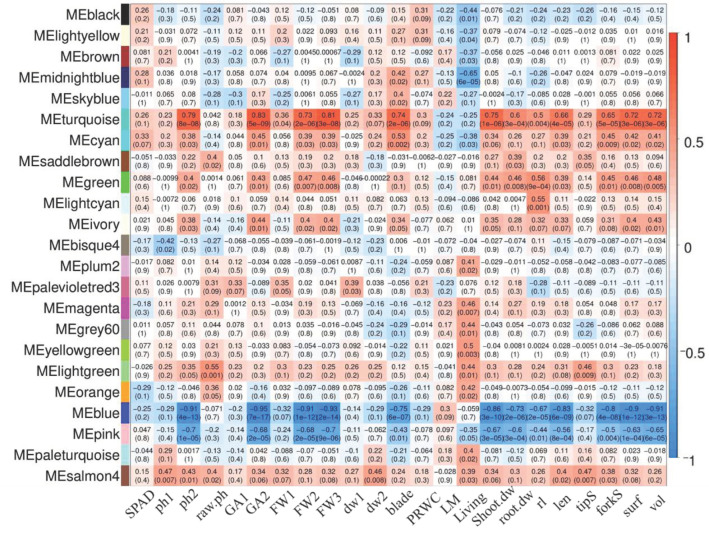
Module-trait relationship between different gene modules. The numbers within the heatmap represent correlations and *p*-value (red, positively correlated; blue, negatively correlated) for the module-trait associations. *X*-axis, phenotypes from left to right are as follows: SPAD, content of relative chlorophyll; ph, plant height; raw.ph, initial plant height; GA, growth amount; FW, fresh weight in above-ground part of maize; dw, dry weight in above-ground part of maize; blade, number of leaves; PRWC, relative water content of plant; LM, leaves mortality; living, survival rate; shoot.dw, dry weight of shoot; root.dw, dry weight of root; rl, primary root length; len, root length; tipS, number of root tips; forkS, number of root forks; surf, root surface area; vol, root volume. “1, 2, and 3” at the end of the parameter names denote different places and years.

**Figure 7 ijms-24-15039-f007:**
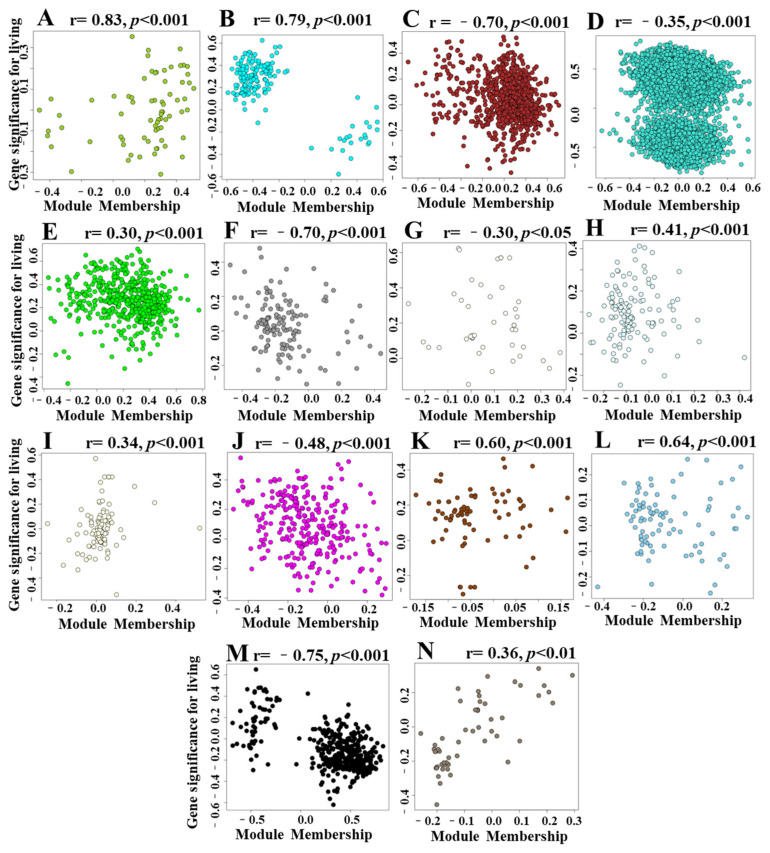
Correlations between gene module memberships (MM) and gene significance (GS) associated with survival rate under drought. (**A**–**N**): MEyellowgreen, MEcyan, MEbrown, MEturquoise, MEgreen, MEgrey60, MEivory, MElightcyan, MElightyellow, MEmagenta, MEsaddlebrown, MEskyblue, MEblack, and MEbisque4.

**Figure 8 ijms-24-15039-f008:**
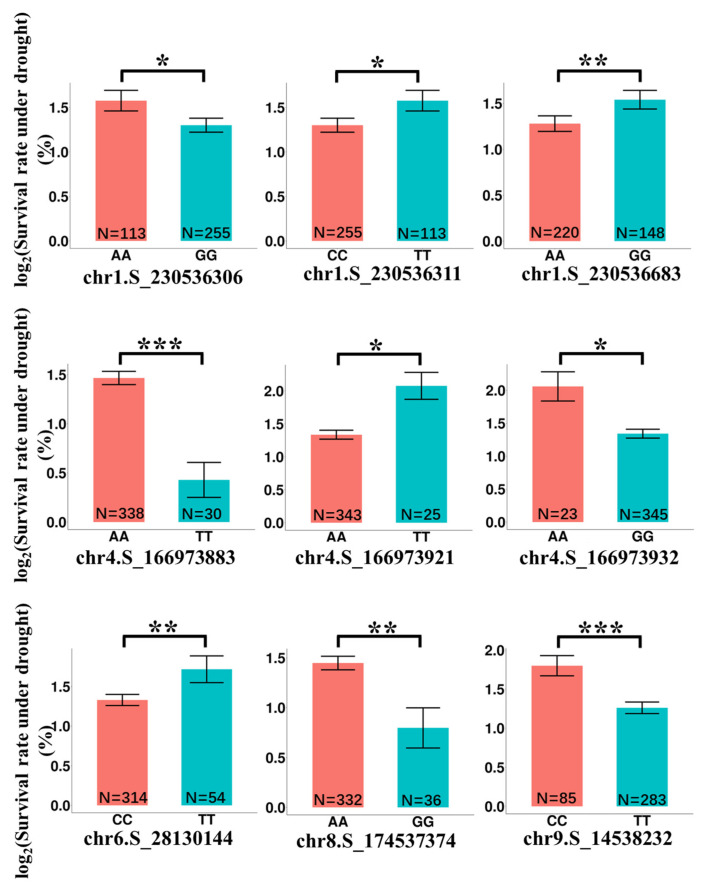
Survival rates differences among different SNP genotypes in natural populations. *, *p* < 0.05; **, *p* < 0.01; ***, *p* < 0.001.

**Figure 9 ijms-24-15039-f009:**
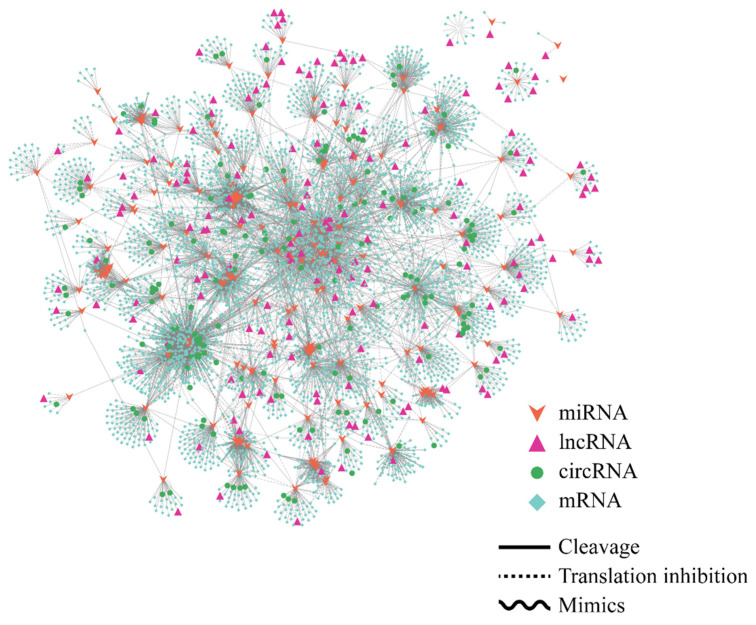
Network of the lncRNA-miRNA-circRNA/mRNA. The arrow nodes represent miRNAs. The triangle nodes represent lncRNAs. The round nodes represent circRNAs. The square nodes represent mRNAs. The straight line represents the cleavage function. The dotted line represents the translation inhibition function. The wave line represents the mimics function.

**Table 1 ijms-24-15039-t001:** Numbers and proportions of differentially expressed coding genes and lncRNAs responded to drought stress.

	Total *	Sign (α = 0.05)	Coding Genes	LncRNA Genes	χ^2^ Test (cd–ncd)
Up	107,350	63,005	60,196 (21.4%)	2809 (38.9%)	*p* < 0.001
Down	139,224	87,181	83,537 (29.8%)	3644 (50.5%)	*p* < 0.001
Equal	432,610	137,687	136,920 (48.8%)	767 (10.6%)	*p* < 0.001
Total *	679,184	287,873	280,653	7220	
χ^2^-test (Up–Down)			*p* < 0.001	*p* < 0.001	

*: represents the number of differentially expressed genes in all materials. Up: Up-regulated genes in drought stress; Down: Down-regulated genes in drought stress; Equal: no significantly difference genes in expression levels.

## Data Availability

The transcriptome sequencing data were deposited in the National Genomics DATA Center (NGDC, https://ngdc.cncb.ac.cn/gsub/), the project ID PRJCA019540, smRNA sequencing data (PRJNA816639, PRJNA294848), DNA and RNA methylation of maize AC7643, as well as histone modifications and open chromatin regions identified using ATAC-seq (PRJCA018969). MeDIP sequencing data (SRP063383).

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
