# Peer review of "Identification and Functional Analysis of Drought-Responsive Long Noncoding RNAs in Maize Roots"

_ijms, 2023, doi:10.3390/ijms242015039_

Round 1

Reviewer 1 Report

Tang and collaborators in this worked present a comprehensive approach to identify lncRNAs involved in drought responsiveness is maize focusing on different aspects of transcriptional regulations and building interesting transcriptional networks. The topic, the analyses and the outcome resulting is certainly interesting and worth of publication although, as in many genome-wide manuscripts, only numbers are given and no names. 

Before acceptance there are some issues the authors need to address. The most general one refers to the data described in the manuscript: for some (miRNA, circRNA, phenotypic data) there is no description about the origin, and similarly other info present in the text (for instance, strand specificity) have to be included and described in the Method section.

Very important: you have to make available the data you generated and used in this current paper.

Besides that, I have some questions to address, the first being the expression ratio of the genes for being considered as lncRNA. I read that you regarded as bona fide lncRNA those genes having, among other parameters, an expression level <0.1: is this correct? If this is the case, I'd like to know why the choice of such value, that is not so clear to me. If you apply this threshold for transcripts identified by RNAseq but mapped using uniquely RNCseq you should write it.

The in silico analyses you performed are quite comprehensive, but the presence of data (pictures, RWC measurements, etc) to assess the different behavior of the samples to drought stress would strenghten your results: have you got any? You use phenotypical data (especially for Fig. 6) but no info or refs are provided even if they seem to derive from other publications and therefore not directly applicable in this context.

In general most of the comparisons (differential expressions, lncRNA identified) have been made considering water stress-resistant accessions together with water-sensitive ones; it would be interesting to assess them separately, in order to see whether there are distinctive genes/lncRNAs according to the genetic background. In Figure 3C something has been attempted but it is not evident to me a distinctive separation between the two phenotypes (Drought-tolerant, including both line AC7643 and related tolerant RILs in one side and line AC7729/TZSRW and related sensitive ones).

Some qPCR would be useful to confirm the quality of the sequencing although I'm aware that these may not be directly comparable with RNCseq results.

Related to this: in the current manuscript you are using data (phenotypic, genomic, small RNAs) but you did not indicate (neither in the Method section nor as reference) where such data come from: you have to.

Other remarks are inline in the attached file. Other than those, please note that Figure 1 is repeated 4 times in the text and most of the links to Figures are just showing an error. Figure 6 contains phenotypic data that are not adequately described. Supplementary Table 3: please clarify what "signed" means

I suggest to revise the English language, as some sentences are not clear (eg lines 155-156, 157, 171-173, 408, 601-602, 652, etc and some verbs to be corrected) and need to be re-checked. 

Author Response

Dear Editor,

Thank you very much for giving us the opportunity to submit our revised manuscript. We really appreciate the time and effort invested by you and the reviewers in evaluating our manuscript. We have carefully considered your feedback and have incorporated most of the suggestions made by the reviewers. Additional experiments and analyses have been conducted accordingly to fully address the reviewers’ concerns. The changes are highlighted within the revised manuscript. The point-by-point response to the reviewers’ comments and concerns are as followings. All page numbers refer to the revised manuscript are in a clean version.

Reviewers' comments:

Reviwer 1:

1.Tang and collaborators in this worked present a comprehensive approach to identify lncRNAs involved in drought responsiveness is maize focusing on different aspects of transcriptional regulations and building interesting transcriptional networks. The topic, the analyses and the outcome resulting is certainly interesting and worth of publication although, as in many genome-wide manuscripts, only numbers are given and no names.

Before acceptance there are some issues the authors need to address. The most general one refers to the data described in the manuscript: for some (miRNA, circRNA, phenotypic data) there is no description about the origin, and similarly other info present in the text (for instance, strand specificity) have to be included and described in the Method section.

Very important: you have to make available the data you generated and used in this current paper.

Author response: We appreciate your positive evaluation of our manuscript. The samples were chosen from the recombinant inbred lines (RILs) constructed by the drought tolerant inbred line AC7643 and drought sensitive inbred line AC7729/TZSRW previously. We have several publications based on the RILs [1-3]. The leaves death rate under drought were investigated [1] and 14 RILs, as well as 2 parental inbred lines with distinct drought characteristics were selected for the following experiments [1-3]. The inbred lines used in our manuscript are the same as our previous publication focus on the drought responsive miRNAs [2]. The plants at the five-leaf stage were subjected to a 20% PEG treatment to simulate drought stress. The detailed methods for libraries preparation have been described in our previous research Xu et al.[3]. We have included the detailed descriptions in the methods section of the revised manuscript line 582-603.

We have uploaded the data generated in this study (including the sequencing of rRNA minus libraries and ribosome associated RNAs) to the National Genomics Data Center (NGDC) under the project ID PRJCA019540. This data will be made publicly available upon acceptance of our manuscript. In addition, we have added the project ID of the other publicly available data we used in the Data Availability Statement part (Line 724-725). The genotypes of the RIL population and the maize natural population with 368 different maize inbred lines are provided by Prof. Xuecai Zhang [4](downloaded from The iPlant Collaborative https://cyverse.org/) and Prof. Jianbing Yan [5] (downloaded from http://www.maizego.org/), respectively. We have added the references in line 705-713 of revised manuscript.

  1. Besides that, I have some questions to address, the first being the expression ratio of the genes for being considered as lncRNA. I read that you regarded as bona fide lncRNA those genes having, among other parameters, an expression level <0.1: is this correct? If this is the case, I'd like to know why the choice of such value, that is not so clear to me. If you apply this threshold for transcripts identified by RNAseq but mapped using uniquely RNCseq you should write it.

Author response: We apologize for not providing a clear explanation of the data processing of RNC-seq. The RNC-seq detects only the translating RNAs (associated with ribosomes). In this study, the inclusion of RNC-seq data aims to filter out the protein-coding transcripts. The analysis of RNC-seq is referred to the previous researches [6, 7]. Generally, transcripts with an expression threshold of FPKM > 0.1 are retained in the RNA-seq analysis. Here, transcripts with an expression threshold FPKM > 0.1 in the RNC-seq are considered protein-coding transcripts. Transcripts with FPKM <0.1 in the RNC-seq are indicative of noncoding transcripts, which align with our research objectives. We have added detailed information on RNC-seq processing in our revised manuscript Line 144-151 and Line 606-608.

3.The in silico analyses you performed are quite comprehensive, but the presence of data (pictures, RWC measurements, etc) to assess the different behavior of the samples to drought stress would strenghten your results: have you got any? You use phenotypical data (especially for Fig. 6) but no info or refs are provided even if they seem to derive from other publications and therefore not directly applicable in this context.

Author response: Thank you for your suggestion. Yes, it is quite essential to illustrate the distinct behaviors of the samples. The leaves death rate under drought were investigated previously as shown in the Supplementary S1 of published paper by Tang et al. [2] (as in the following Figure 1). We have added the detailed description and corresponding reference in the method part of our revised manuscript line 582-586.

The phenotypes utilized for WGCNA network construction were obtained as follows: Both the RILs and the parental lines were grown in 30 cm × 30 cm pots, with three seedlings per pot, planted in a mixture of sandy soil (nutrient soil:sand ratio of 1:3). Each inbred line was represented by ten pots, with five pots allocated for drought stress treatment and five for the control group (well water). The treatment group ceased watering when the plants reached the three-leaf stage, while the control group received regular watering. After 40 days of drought treatment, we collected the phenotypes, including aboveground fresh/dry weight, plant height, Soil Plant Analysis Development (SPAD, to estimate the chlorophyll content), leaf mortality rate, and plant survival rate. The experiment was conducted with two replicates. Detailed methods are provided in the revised manuscript, specifically in lines 715-723.

We utilized the phenotypic data, as described above, to construct the WGCNA network. Additionally, GWAS analysis was performed using phenotypic data from the previous studies [8]. The detailed information is also added in the revised manuscript, specifically in lines 705-713.

4.In general most of the comparisons (differential expressions, lncRNA identified) have been made considering water stress-resistant accessions together with water-sensitive ones; it would be interesting to assess them separately, in order to see whether there are distinctive genes/lncRNAs according to the genetic background. In Figure 3C something has been attempted but it is not evident to me a distinctive separation between the two phenotypes (Drought-tolerant, including both line AC7643 and related tolerant RILs in one side and line AC7729/TZSRW and related sensitive ones).

Author response: Thank you so much for the constructive suggestion. To enhance result clarity, we have added the detailed information about differentially expressed lncRNAs into Supplementary Table 2. This will facilitate the selection of distinctive lncRNAs across different genetic backgrounds. 1,145 and 1,140 drought-respond lncRNAs were identified in drought-sensitive and drought-tolerance inbred lines, respectively. There are 833 DEG lncRNAs in different drought tolerant inbred lines (D/S). 312 and 307 of specifically DEG lncRNAs in drought-tolerance and drought-sensitive inbred lines, respectively. We have rewritten the sentences to lines 164-165 and 744-745.

5.Some qPCR would be useful to confirm the quality of the sequencing although I'm aware that these may not be directly comparable with RNCseq results.

Author response: Thank you so much for pointing this out. To address your concerns, we performed qRT-PCR validation on a randomly selected set of five lncRNAs in the RIL parental lines under both well-watered (WW) and water stress (WS) conditions. The results demonstrated a relative high degree of concordance with the ribo-zero sequencing data, specifically confirming the expression of three lncRNAs (XLOC_032450, XLOC_017628, XLOC_028529) as illustrated in Supplementary Figure 1. Notably, for XLOC_007641, while this lncRNA did not exhibit detectable reads in the AC7729/TZSRW inbred line, qRT-PCR detected its expression. Furthermore, XLOC_027965, closely mirrored our predictions in AC7643 (Supplementary Figure 1). The sequences of primers used for qRT-PCR were provided in the Supplemental Table 6. We have integrated the qRT-PCR results into the revised manuscript, specifically within lines 165-173.

Figure 2. qRT-PCR validation of lncRNA expression

The left panel depicts the relative expression levels of each lncRNA in both maize drought-tolerant inbred line AC7643 and drought-sensitive inbred line AC7729/TZSRW under well-watered (WW) and water-stressed (WS) conditions, while the right panel shows the Fragments Per Kilobase of transcript per Million mapped reads (FPKM) values of the lncRNAs in the corresponding ribo-zero libraries. (A) XLOC_007641; (B)XLOC_032450; (C)XLOC_017628; (D) XLOC_027965; (E) XLOC_028529.

6.Related to this: in the current manuscript you are using data (phenotypic, genomic, small RNAs) but you did not indicate (neither in the Method section nor as reference) where such data come from: you have to.

Author response: As indicated in our responses to Question 1# and 3#, we have incorporated the detailed descriptions into the revised manuscript within the specified lines 705-713 and 715-723.

7.Other remarks are inline in the attached file. Other than those, please note that Figure 1 is repeated 4 times in the text and most of the links to Figures are just showing an error. Figure 6 contains phenotypic data that are not adequately described. Supplementary Table 3: please clarify what "signed" means

Author response: Greatly appreciate your meticulous review and attention to detail in checking and improving our manuscript. The responses to the remarks are listed in 8#.

There was an issue with the conversion during the manuscript submission process. We are sorry for the overlooked the conversion errors. We will submit the revised manuscript in PDF format to ensure its correctness. Regarding the clarification in supplementary Table 3, where "signed" means "significance," and "no signed" means "no significance,". We added it in the revised supplementary Table 5.

  1. Other remarks in the manuscript.

Author response: Greatly appreciate your meticulous review and attention to detail in checking and improving our manuscript. We have thoroughly reviewed and addressed each of the questions you raised. All necessary changes and corrections have been made in the revised manuscript.

8.1 the structure of lncRNAs is similar to the most typical mRNAs, with 3′ polyadenylated (polyA+ or polyA-) and 5′capped structures added after their splicing. The “after” is right?

Author response: We apologies for any confusion in our previous description. To clarify, we have made a modification in the manuscript by changing 'after' to 'before' in Line 43 to accurately convey that 5’ capping and 3’ tailing are performed after the formation of pre-mRNA.

8.2 reads were aligned to the merged gtf or singularly to any gtf created from each library?

Author response: Sorry for the confusion. The merged GTF file was generated from the GTF files of the different 32 samples using Stringtie (1.3.5), and all subsequent analyses are taken this reconstructed merged GTF as reference. The RNC-seq reads were also aligned with the merged GTF file. We have updated the methods section in the revised manuscript, specifically in lines 144-151, 606-608 and 651-656.

8.3 please indicate what you mean with respect to the >200k transcripts indicated above

Author response: Thank you so much for pointing this out. In this study, the dataset comprising ">200k transcripts" was generated from the ribo-zero transcriptome data obtained from 32 different samples. Initially, reads from each sample were individually aligned to the maize B73 reference genome (AGPV4) to generate separate GTF files. Subsequently, we reconstructed a merged GTF file by combining data from all the samples using Cuffmerge. This reconstructed transcriptome encompasses a total of 42,427 genes and 201,386 transcripts. In comparison, the annotations of maize AGPV4 reference genome obtained from the MaizeGDB Genome Center, includes 44,117 genes and a total of 135,446 transcripts, encompassing miRNAs and protein-coding genes.

8.4 these info are not shown in any Figure or Table but I think it would be important to

be provided, as Supplementary maybe. Moreover, you made a all-vs-all comparison and not among groups (drought-resistant and drought-sensitive RILs and related parents as you indicated in the Method section). How can you assess the relatedness of a lncRNA to drought if only present in one member of the group?

Author response: Thanks for the constructive comment. As suggested, we have added the detailed information of the drought responsive lncRNAs identified in the Supplemental Table 2. Yes, we performed a comprehensive all-vs-all comparison, but we also delved further into additional comparisons, including drought-resistant vs drought-sensitive and Parents vs RILs, in section 2.3 titled 'LncRNAs in maize roots respond to drought stress.'

8.5 this is not evidenced in Figure 1; it would be better to add an asterisk and relative caption to show it

Author response: Thank you very much for pointing this out. We have added the asterisk to indicate significance in the calculation in Figure 1 of the revised manuscript, as detailed described in lines 185-196.

8.6 it is not clear which numbers the dots represent. If the axis goes from 0 to 9 it seems that there are different samples having the same numbers but in different columns.

Author response: Thank you for your reminder, and we apologize for any confusion caused by our previous descriptions. We have revised the information related to Figure 1A. This figure consists of two histograms and a dot matrix diagram. The left histogram represents the total number of lncRNAs detected in each inbred line, while the upper histogram represents the number of specifically expressed lncRNAs in the inbred lines where the black dots are located. These changes have been made in the revised manuscript, as detailed in lines 187-190.

8.7 This is the case both for down-and up-regulated transcripts? How do you comment this?

Author response: Yes, typically, more genes were down-regulated in the abiotic stress condition. The predominance of down-regulated genes under drought can be attributed to the plant's adaptive response mechanisms. Plants often reduce the expression of non-essential genes to conserve energy and resources for survival under stressed condition. Conversely, up-regulation genes typically involves in the plant's defense and adaptation. As shown in Table 1, the down-regulation rate is significantly higher than the up-regulation rate in both lncRNAs and coding genes. Consequently, the overall expression level is down-regulated. We have added this comment in the Discussion part in lines 513-521

8.8 You should add absolute numbers as well to allow readers to link this to Figure 2. The differences refer to RILs only and the significance is related to total numbers, right? Did you check whether specific genes were drought-related? Those may be unrelated to that phenotype

Author response: Thank you so much for pointing this out. We added the number in the revised manuscript Line 209. Yes, the differences refer to RILs only and the significance is related to total numbers. As suggested, we examined the expression fold changes of sample-specific genes under drought condition, and we observed no significant differences among different inbred lines.

8.9 did you separate the specifically expressed genes between D and S? Were there differences? Similarly for treatment and for those offspring-specific.

Author response: Thank you for pointing this out and that is interesting. Based on your suggestion, we further analyzed the up-regulated and down-regulated of specifically expressed lncRNAs in D/S (as shown in Table 1). It was found that the number of up-regulated specifically expressed lncRNAs in drought-tolerance inbred lines (D group) was significantly lower than that of in drought-sensitive inbred lines (S group). While the number of down-regulated specifically expressed lncRNAs was significantly in D higher than that of in S. We have added it in the result part in line 290-296. LncRNAs in D and S that specifically respond to stress are enriched into the same module named MEgrey60. We extracted the coding genes in the same co-expression network modules (significantly correlated with plant survival rate under stress) with specifically expressed lncRNAs. GO enrichment analysis was performed on these coding genes to reveal that those genes are enriched in "GO:0044765 single-organism transport" and "GO:1902578 single-organism localization" (Figure 3).

Table 1 Numbers and proportions of specifically expressed lncRNAs in drought-tolerant inbred lines and drought-sensitive inbred lines.

Drought-tolerant inbred lines (D)

Drought-sensitive inbred linesS

c2 -test

(D-vs-S)

Up

184 (7.3%)

290 (11.8%)

p < 0.001

Down

245 (9.8%)

164 (6.7%)

p < 0.001

Total*

2496

2456

*: represents the number of specifically expressed lncRNAs in any of the drought-tolerant/drought sensitive inbred lines. Up: number of up-regulated lncRNAs; Down: number of down-regulated lncRNAs.

Figure 3. GO enrichment analysis of the coding gene in the module of MEgrey60.

8.10 you mean that the sequences encompassing the recombination events were identified 310 lncRNA and 6092 coding genes? In the paper you mentioned, you used only 2 RILs, so this mean that you mapped all the RNAseq (or MSCseq data) of the 16 recombinant inbred lines onto the sequences identified in 2021?

Author response: Sorry for the confusion. Yes, the recombination events in RILs were caculated using the Genotyping-by-sequencing data of the entire RIL population, employing the TASSEL-GBS pipeline. To explore the origin of lncRNA specifically expressed in the offspring, we explored the lncRNA is derived from recombination by detecting the overlap of lncRNAs genes and the recombination site. The detailed method for this analysis can be found in reference [1] and we have provided it in our revised manuscript, specifically in line 230-232.

8.11 the groups are not very visible: please consider not to draw Venn diagrams on scale

Author response: Thank you so much for the suggestion. We have modified the scale of the venn diagram in the revised manuscript Figure 2, based on the suggestion provided.

8.12 what are these numbers referring to? in section 2.1 you stated that 199,536 transcripts have been identified in total, corresponding to 40k genes, et

Author response: Sorry for the confusion. These numbers in Table 1 represent the number of differentially expressed genes in all materials. We have added the corresponding annotation in the revised manuscript line 298.

In section 2.1, as in the case of #8.3, yes, the transcript number is substantial, but it is justified and reasonable, as the maize reference genome annotation is also includes 44,117 genes and a total of 135,446 transcripts.

8.13this is interesting but you should add something in this regard in the Methods section

Author response: Thank you very much for your approval, and we apologize for the missing method description. We utilized the Genome-Wide Association Study (GWAS) using the plant survival rate under drought [9] and kernel oil content from published studies [5]. SNPs that exhibited significant associations with survival rate and kernel oil content were filtered and then the SNP numbers overlaped within the lncRNA locus and protein-coding genes were caculated to evaluate the potential functions of lncRNAs and genes. We have now included a detailed description of this method in the revised manuscript, in lines 708-713.

8.14 in panel A it seems that the opposite is shown (coding - noncoding)

Author response: Thank you so much for pointing this out. We are sorry about the error in Panel A where the coding and noncoding genes are presented in reverse. We have made the correction in the revised manuscript, as indicated in line 328.

8.15 where these numbers come from?

Author response: Sorry for the confusion brought by the typo error. Actually, these numbers represent the differentially expressed lncRNAs and protein-coding genes under drought identified in our ribo-zero sequencing data. The corrected numbers and their context can be found in lines 148-151 of the revised manuscript.

8.16 what do you mean with "adsorb"?

Author response: Apologies for any confusion. In the functional study of circRNAs, we mentioned "adsorb" to describe the ability of circRNA to bind miRNAs, acting as miRNA sponge. Similarly, when studying lncRNAs, which can also bind miRNAs, and we used the term "adsorb" to describe this function. We have modified the descriptions in the revised manuscript, specifically in lines 457and 459.

.

8.17 these are absolutely not visible in the Figure even at high magnification: please provide an higher resolution image.

Author response: Thank you for your feedback. We have now included a higher resolution image in the revised manuscript.

.

8.18 H3K9me3 it is a repressor of transcription, as you correctly indicated in the introduction (lines 114-115), and in fact it is more enriched in lncRNAs than in coding genes (check Fig. 4B). H3K9me3 is higher in lncRNA (Fig 4B).

Author response: Thank you so much for your meticulous review and for pointing out our mistakes regarding H3K9me3. We sincerely apologize for the oversight. However, it is worth noting that, based on the scale of ordinates on the Y-axis in Figure 4B, the difference in the level of H3K9me3 enrichment between lncRNAs and coding genes is not statistically significant. This suggests that H3K9me3 may not exert a strong transcriptional inhibitory effect on lncRNAs. We have corrected these errors in the revised manuscript, as reflected in line 345-348.

8.19 what do you mean with "material"?

Author response: Sorry for the confusion. In our study, the term “material” refers to the maize inbred lines.

8.20 this is correct and a nice example: did you check whether, you found NCED3 ortholog or any other gene known to be responsive to drought among your DEGs? And, in case, did you check the related module/network? This could strenghten your results and give further hints about networks related to drought. qPCR could be useful for valiation. Moreover, it would have been useful an, albeit raw, characterization of DEGs in relation to their putative function. If I'm not wrong, you did mix together both DEGs from D and S, preventing from identifying those lncRNAs (and coding genes) with a

protective role vs drought.

Author response: Thanks for the constructive comment. As you said, we found 6 NCED3 homologous genes (Zm00001d041319, Zm00001d013689, Zm00001d031086, Zm00001d033222, Zm00001d035724, Zm00001d041319) in maize. We observed distinct gene expression patterns across different materials:

Zm00001d041319: No differential expression was detected in any of the inbred lines.

Zm00001d013689: Differentially expressed in 6 inbred lines, with up-regulation in 2 samples and down-regulation in 4 samples.

Zm00001d031086: Only up-regulated in 2 inbred lines.

Zm00001d033222: Differentially expressed in 9 inbred lines, with up-regulation in 7 samples and down-regulation in 2 samples.

Zm00001d035724: Differentially expressed in only one inbred line.

Zm00001d041319: No differential expression was observed in any of the inbred lines.

However, it's worth noting that these genes were not found in any modules associated with drought-related phenotypes

Fortunately, we found a significantly up-regulated WRKY (XLOC_030732) in drought-tolerance inbred line in MEbrown significantly associated with drought survival. WRKY are a kind of transcription factor that has been reported to be response drought. Therefore, the lncRNA located in MEbrown also respond to drought stress. We have added them in the discussion part of the revised manuscript line 545-547.

Based on your suggestion, we further analyzed the up-regulated and down-regulated of specifically expressed lncRNAs in D/S (shown in Table 1). It was found that the number of up-regulated specifically expressed lncRNAs in drought-tolerance inbred lines was significantly lower than that of in drought-sensitive inbred lines. While the number of down-regulated specifically expressed lncRNAs was significantly in D higher than that of in S. We have added it in the result part in line 290-296.

8.21 do those candidates belong to the same module or have interesting target genes identified?

Author response: Thank you for the suggestion. Unfortunately, lncRNAs that harboring those candidates SNPs identified are not in the same module. But we have discovered many transcription factors related to drought stress in the module, such as members belongs to MYB, bZIP, WRKY. A significantly up-regulated transcript from WRKY (XLOC_030732) family was identified in the drought-tolerance inbred lines in the module of MEbrown, which was significantly associated with plant survival rate under drought.

8.22 did you check whether the protective SNP is associated in D or S and the putative

target coding genes?

Author response: Thank you for the suggestion. As suggested, the lncRNAs with the protective SNPs did not show any preference in drought-tolerant or drought-sensitive inbred lines among RILs.

8.23 this is comprehensive of putative intronic and antisense lncRNA, so that you considered only lincRNA in your analyses?

Author response: We regard coding genes that simultaneously transcribe lncRNAs and mRNAs, which is a more rigorous identification process for lncRNAs. It also means that our lncRNAs are lincRNAs.

8.24 how you calculated this? (software)

Author response: Thank you so much for pointing this out. Shannon entropy (R language scripts) was used to evaluate the expression specificity of genes in all the maize lines studied. Shannon entropy was calculated as follows:

where H(X) is the Shannon entropy, x is the gene expressed in each sample, and P(x) is the relative expression in each sample. We have added the detailed descriptions in revised manuscript line 699-703.

References

  1. Wang Q, Xu J, Pu X, Lv H, Liu Y, Ma H, Wu F, Wang Q, Feng X, Liu T, et al: Maize DNA Methylation in Response to Drought Stress Is Involved in Target Gene Expression and Alternative Splicing. Int J Mol Sci 2021, 22.
  2. Tang Q, Lv H, Li Q, Zhang X, Li L, Xu J, Wu F, Wang Q, Feng X, Lu Y: Characteristics of microRNAs and Target Genes in Maize Root under Drought Stress. International Journal of Molecular Sciences 2022, 23.
  3. Xu J, Wang Q, Freeling M, Zhang X, Xu Y, Mao Y, Tang X, Wu F, Lan H, Cao M, et al: Natural antisense transcripts are significantly involved in regulation of drought stress in maize. Nucleic Acids Res 2017, 45:5126-5141.
  4. Zhang X, Perez-Rodriguez P, Semagn K, Beyene Y, Babu R, Lopez-Cruz MA, San Vicente F, Olsen M, Buckler E, Jannink JL, et al: Genomic prediction in biparental tropical maize populations in water-stressed and well-watered environments using low-density and GBS SNPs. Heredity (Edinb) 2015, 114:291-299.
  5. Li H, Peng Z, Yang X, Wang W, Fu J, Wang J, Han Y, Chai Y, Guo T, Yang N, et al: Genome-wide association study dissects the genetic architecture of oil biosynthesis in maize kernels. Nat Genet 2013, 45:43-50.
  6. Luo Z, Hu H, Liu S, Zhang Z, Li Y, Zhou L: Comprehensive analysis of the translatome reveals the relationship between the translational and transcriptional control in high fat diet-induced liver steatosis. RNA Biol 2021, 18:863-874.
  7. Zhang M, Zhao K, Xu X, Yang Y, Yan S, Wei P, Liu H, Xu J, Xiao F, Zhou H, et al: A peptide encoded by circular form of LINC-PINT suppresses oncogenic transcriptional elongation in glioblastoma. Nat Commun 2018, 9:4475.
  8. Yang N, Lu Y, Yang X, Huang J, Zhou Y, Ali F, Wen W, Liu J, Li J, Yan J: Genome wide association studies using a new nonparametric model reveal the genetic architecture of 17 agronomic traits in an enlarged maize association panel. PLoS Genet 2014, 10:e1004573.
  9. Liu S, Li C, Wang H, Wang S, Yang S, Liu X, Yan J, Li B, Beatty M, Zastrow-Hayes G, et al: Mapping regulatory variants controlling gene expression in drought response and tolerance in maize. Genome Biol 2020, 21:163.

Reviewer 2 Report

1) abstract too long - too complicate to understand home take message

2) should be shortened and describve the mode of action of lnRNA, not only listening them

3) method ok, bnut verification of validuty of expression data missing - e.g. qRT PCR,

4) result difficult to read, results nmot verified by qRTPCR

5) discussion - repetion of results, not data based interpretation in some cases-.

Some added value and clear added value of obtained data to curent knowledge would be welcome

english should be improverd

Author Response

Dear Editor,

Thank you very much for giving us the opportunity to submit our revised manuscript. We really appreciate the time and effort invested by you and the reviewers in evaluating our manuscript. We have carefully considered your feedbacks and have incorporated most of the suggestions made by the reviewers. Additional experiments and analyses have been conducted accordingly to fully address the reviewers’ concerns. The changes are highlighted within the revised manuscript. The point-by-point response to the reviewers’ comments and concerns are as followings. All page numbers refer to the revised manuscript are in a clean version.

Reviewers' comments:

Reviewer2:

  • abstract too long - too complicate

Author response: Sorry for the confusion. We deleted relatively inconspicuous conclusions and simplified the statement structure. Please allow us to briefly elaborate the points of the abstract. Firstly, we identify the lncRNA from different inbred lines and conditions. Secondly, comparing the differences in length and number of exons between lncRNA and coding gene. Thirdly, lncRNAs were more susceptible to drought induction. In addition, lncRNA exhibits abundant epigenetic modifications. Lastly, 13 modules (associated with drought) identified through co-expression network analysis, and 9 SNPs located in lncRNA correlated with plant survival rate under drought. The construction of ceRNA networks reveals that lncRNAs can potentially regulate miRNAs and target transcripts in response to drought. The abstract was modified in the revised manuscript.

  • should be shortened and describve the mode of action of lnRNA, not only listening them.

Author response: Sorry for the confusion and thank you so much for pointing this out. In this study, we conducted a comprehensive analysis of lncRNAs, including an investigation into their sequence characteristics, expression patterns, response patterns, and levels of epigenetic modification, using a substantial amount of high-throughput sequencing data. Actually, we performed a comprehensive all-vs-all comparison and the additional lncRNA expressional comparisons, including drought-resistant vs drought-sensitive and Parents vs RILs, in section 2.3 titled 'LncRNAs in maize roots respond to drought stress.'

  • method ok, bnut verification of validuty of expression data missing - e.g. qRT PCR,

Author response: Thank you so much for pointing this out. To address your concerns, we performed qRT-PCR validation on a randomly selected set of five lncRNAs in the RIL parental lines under both well-watered (WW) and water stress (WS) conditions. The results demonstrated a relative high degree of concordance with the ribo-zero sequencing data, specifically confirming the expression of three lncRNAs (XLOC_032450, XLOC_017628, XLOC_028529) as illustrated in Supplementary Figure 1. Notably, for XLOC_007641, while this lncRNA did not exhibit detectable reads in the AC7729/TZSRW inbred line, qRT-PCR detected its expression. Furthermore, XLOC_027965, closely mirrored our predictions in AC7643 (Supplementary Figure 1). The sequences of primers used for qRT-PCR were provided in the Supplemental Table 6. We have integrated the qRT-PCR results into the revised manuscript, specifically within lines 165-173.

Figure 2. qRT-PCR validation of lncRNA expression

The left panel depicts the relative expression levels of each lncRNA in both maize drought-tolerant inbred line AC7643 and drought-sensitive inbred line AC7729/TZSRW under well-watered (WW) and water-stressed (WS) conditions, while the right panel shows the Fragments Per Kilobase of transcript per Million mapped reads (FPKM) values of the lncRNAs in the corresponding ribo-zero libraries. (A) XLOC_007641; (B)XLOC_032450; (C)XLOC_017628; (D) XLOC_027965; (E) XLOC_028529.

  • result difficult to read, results nmot verified by qRTPCR

Author response: Sorry for the confusion. As suggested, qRT-PCR have been performed to validate our results as above.

5) discussion - repetion of results, not data based interpretation in some cases-

Author response: Thank you so much for pointing this out. We have rewritten the discussion section in the revised manuscript.

  1. Other opinion involved in the manuscript.

Author response: Greatly appreciate your meticulous review and attention to detail in checking and improving our manuscript. We have thoroughly reviewed and addressed each of the questions you raised. All necessary changes and corrections have been made in the revised manuscript.

6.1 two categories ?????

Author response: Sorry for the confusion. To explore the relationship between sequence characteristics and drought, we conducted a comparison between drought-responsive and non-responsive lncRNAs based on gene length and exon count.

6.2  The expression of lncRNAs is usually very low and has a specific pattern。Under which condition?

Author response: Sorry for the confusion. Yes, lncRNAs are usually expressed at low levels, lack conservation among species and often exhibit tissue-specific/cell-specific expression patterns [1-3]. We find drought-responsive lncRNAs had different sequence characteristics in length of genes and number of exons, compared with non-responsive lncRNAs. The ratio of down-regulated lncRNAs induced by drought was significantly higher than that of coding genes.

6.3 There are 8,449 drought-responsive transcripts and 1,724 lncRNAs identified in maize and 664 lncRNAs for drought response. Identified Up to now?

Author response: Apologies for any confusion. The numbers of drought-responsive transcripts and lncRNAs were obtained from the respective reference, where they had been identified in maize B73 leaves. We have rephrased these sentences in the revised manuscript for clarity.

6.4 Why choose the 0.1 as the threshold to identify the lncRNA.

Author response: We apologize for not providing a clear explanation of the data processing of RNC-seq. The RNC-seq detects only the translating RNAs (associated with ribosomes). In this study, the inclusion of RNC-seq data aims to filter out the protein-coding transcripts. The analysis of RNC-seq is referred to the previous researches [4, 5]. Generally, transcripts with an expression threshold of FPKM > 0.1 are retained in the RNA-seq analysis. Here, transcripts with an expression threshold FPKM > 0.1 in the RNC-seq are considered protein-coding transcripts. Transcripts with FPKM <0.1 in the RNC-seq are indicative of noncoding transcripts, which align with our research objectives. We have added detailed information on RNC-seq processing in our revised manuscript Line 144-151 and Line 606-608.

6.5 These transcripts originated from 42,427 genes, including 1,923 lncRNA genes and 40,504 coding genes.

Author response: Thank you so much for pointing this out. In this study, the dataset comprising ">200k transcripts" was generated from the ribo-zero transcriptome data obtained from 32 different samples. Initially, reads from each sample were individually aligned to the maize B73 reference genome (AGPV4) to generate separate GTF files. Subsequently, we reconstructed a merged GTF file by combining data from all the samples using Cuffmerge. This reconstructed transcriptome encompasses a total of 42,427 genes and 201,386 transcripts. In comparison, the annotations of maize AGPV4 reference genome obtained from the MaizeGDB Genome Center, includes 44,117 genes and a total of 135,446 transcripts, encompassing miRNAs and protein-coding genes.

6.6 Previous studies utilized Shannon entropy to evaluate the specificity of gene expression among different maize inbred lines [7, 54]. We followed this method and found that lncRNA genes exhibited stronger material-specificity than coding genes. add here brief explasnation.

Author response: Thank you so much for pointing this out. Shannon entropy (R language scripts) was used to evaluate the expression specificity of genes in all the maize lines studied. Shannon entropy was calculated as follows:

where H(X) is the Shannon entropy, x is the gene expressed in each sample, and P(x) is the relative expression in each sample. We have added the detailed descriptions in revised manuscript line 699-703.

6.7 This suggests that recombination events at specific loci more likely to express new

lncRNAs. Why?

Author response: To investigate the origin of lncRNAs specifically expressed in the offspring, we explored whether lncRNAs are derived from recombination events by examining the overlap with recombination sites. We utilized Genotyping-by-sequencing data to calculate recombination sites. Our findings revealed that the proportion of lncRNAs specifically expressed in the Recombinant Inbred Lines (RILs) within recombinant sites was higher than that of RILs not located in recombinant sites and also higher than that of coding genes. Based on this, we conclude that a greater number of specifically expressed lncRNAs are derived from recombination sites. In response to your feedback, we have revised the description in the manuscript, specifically in lines 527-539, to provide further clarity.

6.8 Although lncRNA cannot encode a protein, it can respond to drought stress through various regulatory pathways. mostly repetion of results.

Author response: Thank you for your reminder, and we apologize for any shortcomings in our previous descriptions. Following your valuable feedback, we have revised the discussion section in the revised manuscript in lines 485-521.

6.9 5 mg of total RNA without rRNA was broken into approximately 100bp frag- 606

ments using a Qsonica Q800R3 sonicator (Qsonica LLC, Newton, USA). Or used?

Author response: Thank you so much for pointing this out. We have made the necessary modifications in the revised manuscript. In our protocol, 5 mg of total RNA was fragmented into approximately 100 bp using a Qsonica Q800R3.

References

  1. Cabili MN, Trapnell C, Goff L, Koziol M, Tazon-Vega B, Regev A, Rinn JL: Integrative annotation of human large intergenic noncoding RNAs reveals global properties and specific subclasses. Genes Dev 2011, 25:1915-1927.
  2. Derrien T, Johnson R, Bussotti G, Tanzer A, Djebali S, Tilgner H, Guernec G, Martin D, Merkel A, Knowles DG, et al: The GENCODE v7 catalog of human long noncoding RNAs: analysis of their gene structure, evolution, and expression. Genome Res 2012, 22:1775-1789.
  3. Li L, Eichten SR, Shimizu R, Petsch K, Yeh CT, Wu W, Chettoor AM, Givan SA, Cole RA, Fowler JE, et al: Genome-wide discovery and characterization of maize long non-coding RNAs. Genome Biol 2014, 15:R40.
  4. Luo Z, Hu H, Liu S, Zhang Z, Li Y, Zhou L: Comprehensive analysis of the translatome reveals the relationship between the translational and transcriptional control in high fat diet-induced liver steatosis. RNA Biol 2021, 18:863-874.
  5. Zhang M, Zhao K, Xu X, Yang Y, Yan S, Wei P, Liu H, Xu J, Xiao F, Zhou H, et al: A peptide encoded by circular form of LINC-PINT suppresses oncogenic transcriptional elongation in glioblastoma. Nat Commun 2018, 9:4475.

Round 2

Reviewer 1 Report

I thank the authors for having addressed my comments. I appreciate the replies they wrote and I think these are reasonable. I just ask to provide a reference of the FPKM value threshold they used and add in the correct section and to move paragraphs in the Material and Method section according to the order of their presentation in the Results section.

Please check again Fig. 1 as it appears 4 times (and the pictures are cropped) and the incorrect links to the Figures ("Error! Reference source not found." in the main text).

Still some words missing and/or verbs that should be checked (eg.: lines 141-142, 146-147, 148-149, etc), please recheck 

Author Response

Dear Editor,

Thank you very much for dedicating your time and effort to reevaluate our manuscript. We have thoughtfully incorporated your suggestions and made the necessary modifications in the revised version of the manuscript.

Reviewers' comments:

  1. I thank the authors for having addressed my comments. I appreciate the replies they wrote and I think these are reasonable. I just ask to provide a reference of the FPKM value threshold they used and add in the correct section and to move paragraphs in the Material and Method section according to the order of their presentation in the Results section.

Author response: Thank you very much for your approval of our revised manuscript. We have added the corresponding reference [1-3] in the manuscript to explain why 0.1 was used as the threshold for FPKM. We have added the reference in the revised manuscript line 149 and 152. Furthermore, we have relocated the description of the lncRNA recognition process to the Materials and Methods section. Detailed methods can now be found in the revised manuscript, specifically in lines 611-628. Additionally, we have reorganized the Materials and Methods section to align with the order presented in the Results section.

  1. Please check again Fig. 1 as it appears 4 times (and the pictures are cropped) and the incorrect links to the Figures ("Error! Reference source not found." in the main text).

Author response: I'm sorry that this problem still occurred in the previous submission of the manuscript. We will submit the revised manuscript to ensure its correctness in this time.

  1. Still some words missing and/or verbs that should be checked (eg.: lines 141-142, 146-147, 148-149, etc), please recheck 

Author response: Thank you very much for pointing out these errors. We have made corrections in the revised manuscript with track changes.

  1. Luo Z, Hu H, Liu S, Zhang Z, Li Y, Zhou L: Comprehensive analysis of the translatome reveals the relationship between the translational and transcriptional control in high fat diet-induced liver steatosis. RNA Biol 2021, 18(6):863-874.
  2. Zhang M, Zhao K, Xu X, Yang Y, Yan S, Wei P, Liu H, Xu J, Xiao F, Zhou H et al: A peptide encoded by circular form of LINC-PINT suppresses oncogenic transcriptional elongation in glioblastoma. Nat Commun 2018, 9(1):4475.
  3. Li L, Eichten SR, Shimizu R, Petsch K, Yeh CT, Wu W, Chettoor AM, Givan SA, Cole RA, Fowler JE et al: Genome-wide discovery and characterization of maize long non-coding RNAs. Genome biology 2014, 15(2):R40.
